# The Statistical Cost of Robust Kernel Hyperparameter Tuning

**Raphael A. Meyer**
Tandon School of Engineering
New York University
Brooklyn, NY 11201
`ram900@nyu.edu`

**Christopher Musco**
Tandon School of Engineering
New York University
Brooklyn, NY 11201
`cmusco@nyu.edu`

## Abstract

This paper studies the statistical complexity of kernel hyperparameter tuning in the setting of active regression under adversarial noise. We consider the problem of finding the best interpolant from a class of kernels with unknown hyperparameters, assuming only that the noise is square-integrable. We provide finite-sample guarantees for the problem, characterizing how increasing the complexity of the kernel class increases the complexity of learning kernel hyperparameters. For common kernel classes (e.g. squared-exponential kernels with unknown lengthscale), our results show that hyperparameter optimization increases sample complexity by just a logarithmic factor, in comparison to the setting where optimal parameters are known in advance. Our result is based on a subsampling guarantee for linear regression under multiple design matrices which may be of independent interest.

## 1 Introduction

In machine learning, Kernel Ridge Regression (KRR) is central to modern time series analysis and nonparametric regression. For time series, Gaussian Processes model the covariance of a stochastic process using a kernel matrix, and interpolate the underlying signal with KRR [RW06]. In nonparametric regression, kernels define a local-averaging scheme, and KRR provides a smooth interpolation for the function [Fot07, Tsy09]. Experimentally, it is known that the Kernel Ridge Regression estimator generalizes and interpolates well over continuous domains [WA13, AKM$^+$19].

However, it is also known that *kernel regression only performs well when kernel hyperparameters are chosen well* [WN15, BMS$^+$19]. This observation has lead to significant interest in algorithms that try to find the best kernel parameters in a large search space [WPG$^+$19, LCB$^+$04]. Additionally, the existing research generally assumes that observation noise is independent, unbiased, and random [RW06, MRT12, AKM$^+$17, Fot07]. The goal of this paper is to understand the *statistical cost* of this sort of hyperparameter optimization when we can have *worst-case observation noise*. How many data samples are needed to avoid over-fitting when searching over such a large class of models?

We formalize this problem in an adversarial noise setting that originated in the function approximation community [CP19a, CM17]. By Bochner's theorem, every stationary (shift invariant) kernel function $k_\mu$ can be written $k_\mu(\Delta) = \int_{\mathbb{R}} e^{-2\pi i \xi \Delta} \mu(\xi) d\xi$ for some probability density function $\mu$. [AKM$^+$19] introduced the following active regression problem for interpolating with a fixed kernel $k_\mu$:

**Problem 1.** *Let $y(t)$ be a signal we wish to interpolate. Let $z(t)$ be an adversarial noise signal. Fix regularization parameter $\varepsilon > 0$ and observe $y(t) + z(t)$ at any chosen times $t_1, \ldots, t_n$. How large does $n$ need to be so that an interpolant $\tilde{y}$ constructed from our observations satisfies:*

$$\|\tilde{y} - y\|_T^2 \le C_0 \cdot \left( \|z\|_T^2 + \varepsilon \cdot \text{Energy}_\mu(y) \right)$$

*for some universal constant $C_0 > 1$.*

Here $\|x\|_T^2 := \int_0^T |x(t)|^2 \frac{1}{T} dt$ is the natural $\ell_2$ norm on $[0, T]$. Defined formally in Section 3, $\text{Energy}_\mu(y)$ is a natural measure of the cost of representing the ground truth signal $y$ with the kernel $k_\mu$. It is roughly equal to the smallest norm of a signal capable of using $k_\mu$ to exactly reconstruct $y$. Intuitively, if the kernel $k_\mu$ cannot represent $y$ easily, then the associated term $\text{Energy}_\mu(y)$ is large, and hence the interpolation error may be large.

Problem 1 is a robust active nonparametric learning problem. It is nonparametric in the sense that a kernel is being used to interpolate the signal $y$. It is robust in the sense that the noise function $z(t)$ is arbitrary (for instance, we do *not* assume that $z(t)$ is a zero-mean stochastic process). It is active in the sense that the user chooses the time points $t_1, \ldots, t_n$.

[AKM$^+$19] shows that if we let the number of observations $n$ exceed a natural *Statistical Dimension* parameter which is a function of the kernel $k_\mu$ and regularization parameter $\varepsilon$ (see Section 2.2 for a formal definition), then KRR solves Problem 1. Moreover, for many common kernels (square exponential, sinc, Lorentzian, etc.) this number of samples is necessary in the worst-case.

Since the observation noise $z$ is adversarial, a linear dependence on $\|z\|_T^2$ is inevitable. On the other hand, the $\text{Energy}_\mu(y)$ term can be reduced by decreasing $\varepsilon$, but this increases the statistical dimension of the problem, necessitating more samples. Alternatively, we can decrease the energy term substantially by simply choosing a different kernel. This is the problem of kernel hyperparameter tuning:

**Problem 2.** *Let $y(t)$ be a signal we wish to interpolate. Let $z(t)$ be an adversarial noise signal. Let $\mathcal{U}$ be a (possibly infinite) set of kernel PDFs. Fix regularization parameter $\varepsilon > 0$ and observe $y(t) + z(t)$ at any chosen times $t_1, \ldots, t_n$. How large does $n$ need to be so that we can select a PDF $\tilde{\mu} \in \mathcal{U}$ and construct a KRR interpolant $\tilde{y}$ from our observations such that:*

$$\|\tilde{y} - y\|_T^2 \leq C_0 \cdot \left( \|z\|_T^2 + \varepsilon \cdot \min_{\mu \in \mathcal{U}} \text{Energy}_\mu(y) \right)$$

*for some universal constant $C_0$?*

We should think of $\mathcal{U}$ as containing all PDFs corresponding to kernels in a structured class: for examples, all squared exponential kernels with unknown lengthscale. To solve Problem 2, we must find hyperparameters that are competitive with the best possible kernel defined by $\mathcal{U}$. This is still a robust active nonparametric learning problem, but is now generalized to consider hyperparameters. At a high level, our main result is to prove that the number of time samples required to solve Problem 2 is not much larger than the number of samples required to solve Problem 1.

## 1.1 Prior work

There is substantial prior work on hyperparameter tuning between the Learning Theory, Time Series, and Signal Processing literatures. In the Learning Theory community, the problem of "learning kernels" is typical, but usually assumes we are given a finite set of fixed kernels and have to learn how to combine the given kernels [CMR09, ZO07, MH19]. There does exist some work that discusses tuning hyperparameters for kernel families, but these works all make iid noise assumptions [YC10]. There is also work on gradient methods for hyperparameter tuning, but this work generally avoids finite sample complexity bounds [RW06, BMS$^+$19]. In signal processing, kernel hyperparameter tuning generalizes the well studied problem of spectrum-blind signal reconstruction [FB96, Bre08, ME09]. However, prior work in that area again does not provide finite sample complexity bounds. The core technical results of this paper extend tools from recent work on Randomized Signal Processing. In particular, papers in this area deal with adversarial observation noise, but either assume we know the kernel function exactly [AKM$^+$19] or primarily address Fourier sparse function fitting [CKPS16].

We illustrate the application of our results on the Spectral Mixture (SM) Kernel, introduced by [WA13]. The SM Kernel has garnered interested in the Gaussian process community for its ability to interpolate and extrapolate periodic structure [WA13, YWSS15, TBT15], with applications ranging from modeling human decision-making [WDLX15] to estimating the life span of batteries [HSSM15]. While lots of work has shown that SM Kernel hyperaerameter tuning is difficult in practice [WN15, BMS$^+$19, BHH$^+$16, HDS17, WDLX15], these works do not show if this tuning is difficult due to a lack of data, or due to a lack of strong enough algorithms. We resolve this by showing that not much data is needed to tune SM Kernels.

## 1.2 Contributions

Our main contribution is to extend work on Randomized Signal Processing to bound the sample complexity of Problem 2. For many cases (i.e. the Squared Exponential Kernel with unknown lengthscale), we prove that the sample complexity of learning both the hyperparameters and the signal $y(t)$ is only logarithmically larger than the sample complexity of learning $y(t)$ with known hyperparameters (see Corollary 3 with $q = 1$). In other words, solving Problem 2 is not much harder than solving Problem 1 with the hardest single kernel in $\mathcal{U}$. We prove this in two core steps:

- First, we consider the setting where we want to optimize over a large but finite set of $Q$ possible kernels. To solve Problem 1, the problem where we have a fixed kernel, prior work requires the number of samples to depend *linearly* on $1/\delta$ [AKM$^+$19]. Accordingly, a naïve solution to Problem 2 that combines existing results with a union bound would require the number of samples to grow linearly with $Q$. In Section 4 we improve this dependence to be logarithmic. Our result requires a subsampling guarantee for linear operators that may have infinite dimension. When applied to finite matrices, this result corresponds to a guarantee for subsampled linear regression with multiple design matrices.

- Next, we show how to use this result to bound the sample complexity of hyperparameter tuning for kernels with an *infinite space of hyperparameters*. In particular, Section 5 shows how to discretize the space of hyperparameters, reducing the problem from picking a hyperparameter in a continuous space to picking a hyperparameter from a finite set. Then, the result from the first bullet point bounds the actual sample complexity of learning our hyperparameters. For demonstration purposes, a full analysis is presented for the commonly used Spectral Mixture (SM) Kernel, but the broad framework generalizes to most other stationary kernels.

Recall from the discussion on prior work that SM Kernel Hyperparameter tuning is known to be difficult in practice. However, it was not known if this tuning is difficult because SM Kernels require many observations, or if we lack algorithms that can efficiently find a good set of hyperparameters. The second bullet point above resolves this uncertainty under the weak noise assumptions of Problem 2, showing that even without random noise, a small number of observations can statistically identify a near-optimal SM Kernel.

## 2 Preliminaries

Let bold capital letters, like $A$ and $B$, denote complex-valued matrices. Let bold lower case letters, like $\mathbf{x}$ and $\mathbf{y}$, denote complex-valued vectors. $\|\mathbf{x}\|_2$ denotes the $\ell_2$ norm of $\mathbf{x}$. We view infinite-dimensional linear operators as generalization of matrices, and functions as generalizations of vectors, so the notation used with be analogous. Calligraphic capital letters, like $\mathcal{A}$ and $\mathcal{B}$ will represent either linear operators or sets; it will be clear from context. Lower case non-bold letters, like $f$ and $g$, denote complex-valued functions of real numbers. Typically $y(t)$ and $z(t)$ will represent functions in the time domain, while $g(\xi)$ and $h(\xi)$ will represent functions in the frequency domain. We use $\preceq$ and $\succeq$ to denote semidefinite order for both matrices and Hermitian operators.

In general, we use $\mathcal{H}$ to denote a Hilbert space. $\langle f, g \rangle_{\mathcal{H}}$ and $\| \cdot \|_{\mathcal{H}}$ denote the corresponding inner product and norm. For a complex number $x$, we let $x^*$ denote its complex conjugate. For a matrix or linear operator $\mathcal{A}$, we let $\mathcal{A}^*$ denote the Hermitian adjoint. That is, if $\mathcal{A}$ maps between Hilbert spaces $\mathcal{H}_1$ and $\mathcal{H}_2$, then $\mathcal{A}^* : \mathcal{H}_2 \to \mathcal{H}_1$ satisfies $\langle f, \mathcal{A}^* g \rangle_{\mathcal{H}_1} = \langle \mathcal{A}f, g \rangle_{\mathcal{H}_2}$ for any $f \in \mathcal{H}_1, g \in \mathcal{H}_2$.

### 2.1 Shift Invariant Kernels

This paper is concerned with shift-invariant, positive semidefinite kernel functions on the real line. By Bochner's theorem, any such kernel is the Fourier transform of a positive measure [RR07], and for all settings we consider, the measure will be a probability measure with finitely bounded probability density function $\mu$.[1] We denote the corresponding kernel function by $k_\mu$:

$$k_\mu(t_1 - t_2) = \int_{\xi \in \mathbb{R}} e^{-2\pi i (t_1 - t_2)} \mu(\xi) d\xi. \tag{1}$$

For example, when $\mu(\xi) = \frac{1}{\sqrt{2\pi\sigma^2}} e^{-\xi^2/2\sigma^2}$, $k_\mu(\Delta) = e^{-\Delta^2\sigma^2}$ is a squared exponential kernel, also called a radial basis function (RBF) kernel. When $\mu(\xi) = 1/2F$ for $\xi \in [-F, F]$ and 0 elsewhere, $k_\mu(\Delta) = \text{sinc}(F|\Delta|)$ is a sinc kernel.

We let $L_2(\mu)$ denote the space of complex-valued square integrable functions with respect to $\mu$. $L_2(\mu)$ has inner product $\langle g, h \rangle_\mu := \int_\mathbb{R} g(\xi)^* h(\xi)\mu(\xi)d\xi$ and norm $\|g\|_\mu^2 := \langle g, g \rangle_\mu$. We will also refer to $\|g\|_\mu^2$ as the power of $g$ with respect to $\mu$. A function $g$ is in $L_2(\mu)$ if $\|g\|_\mu < \infty$. We let $L_2(T)$ denote the set of complex-valued square integrable functions on $[0, T]$. I.e. $L_2(T)$ has inner product $\langle x, y \rangle_T := \int_0^T x(t)^* y(t) \frac{1}{T} dt$ and norm $\|x\|_T^2 := \langle x, x \rangle_T$. A function $x$ is in $L_2(T)$ if $\|x\|_T < \infty$.

## 2.2 Statistical Dimension and Universal Sampling

As discussed, the sample complexity of interpolating a function $y$ on $[0, T]$ with a *fixed* kernel function $k_\mu$ is characterized by the statistical dimension of that kernel. Before formally defining this quantity, we introduce the integral operator $\mathcal{K}_\mu : L_2(T) \to L_2(T)$

$$[\mathcal{K}_\mu x](t) := \int_0^T k_\mu(s - t)x(s)\frac{1}{T}ds,$$

which is defined for any kernel function $k_\mu$ and time range $[0, T]$. Note that $\mathcal{K}_\mu = \mathcal{F}_\mu^* \mathcal{F}_\mu$ where $\mathcal{F}_\mu$ and $\mathcal{F}_\mu^*$ are the following Fourier transform and inverse Fourier transform operators:

$$\mathcal{F}_\mu : L_2(T) \to L_2(\mu) \qquad\qquad [\mathcal{F}_\mu x](\xi) := \int_0^T x(t)e^{-2\pi i\xi t}\frac{1}{T}dt$$

$$\mathcal{F}_\mu^* : L_2(\mu) \to L_2(T) \qquad\qquad [\mathcal{F}_\mu^* g](t) := \int_\mathbb{R} g(\xi)e^{2\pi i\xi t}\mu(\xi)d\xi$$

**Definition 1** (Statistical Dimension). *For any bounded PDF $\mu$ with corresponding kernel $k_\mu$, time range $[0, T]$, and parameter $\varepsilon > 0$, the statistical dimension $s_{\mu,\varepsilon}$ is defined:*

$$s_{\mu,\varepsilon} := \text{tr}\left(\mathcal{K}_\mu(\mathcal{K}_\mu + \varepsilon\mathcal{I}_T)^{-1}\right),$$

*where $\mathcal{I}_T$ is the identity operator on $L_2(T)$ and $\text{tr}$ is the trace of an operator.*

Refer to [AKM+19] for bounds on the statistical dimension of common kernels. For example, for an RBF kernel with variance $\sigma^2$, $s_{\mu,\varepsilon} = O(\sigma^2 T\sqrt{\log(1/\varepsilon)} + \log(1/\varepsilon))$. For a sinc kernel with bandlimit $F$, $s_{\mu,\varepsilon} = O(FT + \log(1/\varepsilon))$.

[AKM+19] prove that Problem 1 can be solved with a number of samples depending on the statistical dimension $s_{\mu,\varepsilon}$ as long as active samples are drawn from the following distribution over $[0, T]$:

**Definition 2** (Universal Sampling Distribution[2]). *For a parameter $\alpha > 0$, let*

$$\tilde{\tau}_\alpha(t) := \begin{cases} \frac{\alpha}{\min\{t, T-t\}} & t \in [T\frac{1}{\alpha^6}, T(1 - \frac{1}{\alpha^6})] \\ \frac{\alpha^6}{T} & t \in [0, T\frac{1}{\alpha^6}] \cup [T(1 - \frac{1}{\alpha^6}), T] \end{cases}$$

*Note that $\int_0^T \tilde{\tau}_\alpha(t)dt = O(\alpha\log\alpha)$.*

Surprisingly, this distribution works for *any* kernel PDF $\mu$ and $\varepsilon > 0$, as long as $\alpha \geq cs_{\mu,\varepsilon}$ for some universal constant $c > 0$. Specifically, [AKM+19] show that $n = \Omega(s_{\mu,\varepsilon}(\frac{1}{\delta} + \log(s_{\mu,\varepsilon})))$ independent samples drawn from $[0, T]$ with probability proportional to $\tilde{\tau}_\alpha(t)$ suffice to solve Problem 1 with probability $(1 - \delta)$. The result relies on proving that $\tilde{\tau}_\alpha$ forms an upper bound for the *Ridge Leverage Function* of $\mathcal{F}_\mu^*$. Details are discussed in Appendix C.2, specifically Lemma 3 and Lemma 4.

## 2.3 Spectral Mixture Kernels

The core goal of this paper is to bound the sample complexity of learning kernel hyperparameters under adversarial noise. While our techniques can apply to a wide variety of kernel classes, we

illustrate their application with the Spectral Mixture (SM) Kernel introduced by [WA13]. The SM Kernel is defined by having a PDF that is a symmetric mixture of Gaussians.

Formally, let $\mu_{c,\sigma}(\xi) := \frac{1}{\sqrt{2\pi\sigma^2}} e^{-\frac{(\xi-c)^2}{2\sigma^2}}$ denote a Gaussian PDF with mean $c$ and lengthscale $\sigma^2$. Then, let $\mu_{\mathbf{c},\boldsymbol{\sigma},\mathbf{w}}(\xi) := \sum_{j=1}^{q} w_j \mu_{c_j,\sigma_j}(\xi)$ denote a mixture of $q$ Gaussians with weights in $\mathbf{w}$, means in $\mathbf{c}$, and lengthscales in $\boldsymbol{\sigma}$. In the special case that $\mathbf{w}$ is the all ones vector, we omit $\mathbf{w}$ from the subscript: $\mu_{\mathbf{c},\boldsymbol{\sigma}}(\xi)$. The SM Kernel considers the special case of the mixture of Gaussians kernel when the PDF is symmetric: $d\mu_{\mathbf{c},\boldsymbol{\sigma},\mathbf{w}}(-\xi) = d\mu_{\mathbf{c},\boldsymbol{\sigma}\mathbf{w}}(\xi)$, making the kernel function real-valued:

$$k_{\mathbf{c},\boldsymbol{\sigma},\mathbf{w}}(s-t) = \sum_{j=1}^{q} w_j e^{-2\pi^2(s-t)^2\sigma_j^2} \cos(2\pi(s-t)c_j)$$

All our results are stated for the Mixture of Gaussians kernel, so the SM kernel is handled implicitly.

## 3 Technical Overview

At a high level, we are given a possibly infinite set of PDFs over frequencies $\mathcal{U}$ and want to find a specific PDF $\tilde{\mu} \in \mathcal{U}$ such the KRR interpolant using $\tilde{\mu}$ is a good interpolant for the ground truth signal $y(t)$. We only get to observe $y(t)$ through adversarially perturbed samples, and we get to pick those samples to lie anywhere in $[0,T]$. Our main concern is bounding the number of samples needed to identify a near-optimal $\tilde{\mu}$ and its associated interpolant $\tilde{y}$. We formally state Problem 2 below:

**Problem 2 Restated.** *Let $y(t)$ be a signal we want to interpolate. Let $z(t)$ be an adversarial noise signal. Let $\mathcal{U}$ be a (possibly infinite) set of kernel PDFs. Let $\hat{\mathcal{U}} \subseteq \mathcal{U}$ be the subset of PDF capable of representing $y$ exactly[3]. That is, $\hat{\mathcal{U}} := \{\mu \in \mathcal{U} \mid \exists h \in L_2(\mu), y = \mathcal{F}_\mu^* h\}$. Fix regularization parameter $\varepsilon > 0$ and number of observations $n$. Observe $y(t) + z(t)$ at any chosen times $t_1, \dots, t_n$. Using any $\tilde{\mu} \in \mathcal{U}$, construct an interpolant $\tilde{y}$ from our observations such that*

$$\|y - \tilde{y}\|_T^2 \le C_0 \cdot \left( \|z\|_T^2 + \varepsilon \min_{\mu \in \hat{\mathcal{U}}, y = \mathcal{F}_\mu^* h} \|h\|_\mu^2 \right)$$

*for some universal constant $C_0 > 1$.*

Note that, for any $\mu \in \hat{\mathcal{U}}$, we have defined $\text{Energy}_\mu(y)$ to be $\|h\|_\mu^2$, where $y = \mathcal{F}_\mu^* h$. That is, the energy of $y$ under PDF $\mu$ is the norm of the signal whose Inverse Fourier Transform is $y$. Inuitively, if it is difficult to represent $y$ in $L_2(\mu)$, then the energy of $y$ is large.

To make our statistical approach clear, we start by presenting the exact time-sampling and interpolation schemes used in this paper. We need two algorithms for our analysis: the first picks $n$ times samples and builds $Q$ different weighted kernel matrices (one for each of $Q$ different given kernels); the second constructs a KRR interpolant for any given weighted kernel matrix. Note that all kernel matrices are constructed *using the exact same time samples*.

---

**Algorithm 1** Time Point Sampling

**input**: Kernel functions $k_{\mu_1}, \dots k_{\mu_Q}$, non-negative function $p(t)$ on $[0,T]$ with known integral $P = \int_0^T p(t)dt$, number of samples $n$.
**output**: Times $t_1, \dots, t_n \in [0,T]$, weights $v_1, \dots, v_n$, PSD matrices $\boldsymbol{K}_{\mu_1}, \dots, \boldsymbol{K}_{\mu_Q} \in \mathbb{C}^{n \times n}$.
  1: Independently sample $t_1, \dots, t_n$ from $[0,T]$ with probability proportional to $p(t)$.
  2: For $i \in \{1, \dots, n\}$ set $v_i := \sqrt{\frac{P}{n \cdot T \cdot p(t_i)}}$.
  3: For $q \in \{1, \dots, Q\}$ and $i, j \in \{1, \dots, n\}$ set $[\boldsymbol{K}_{\mu_q}]_{i,j} := v_i v_j \cdot k_{\mu_q}(t_i, t_j)$
  4: **return** $t_1, \dots, t_n, v_1, \dots, v_n, \boldsymbol{K}_{\mu_1}, \dots, \boldsymbol{K}_{\mu_Q}$.

---

Think of $p(t)$ as being $\tau_\alpha(t)$ for some $\alpha$. For any particular $\boldsymbol{K}_\mu$, we can compute and evaluate the interpolant $\tilde{y}$ as follows:

**Algorithm 2** Computing the Interpolant

---

**input**: Time points $t_1, \ldots, t_n \in [0, T]$, weights $v_1, \ldots, v_n$, PSD matrix $\boldsymbol{K}_\mu \in \mathbb{C}^{n \times n}$, regularization parameter $\varepsilon > 0$.
**ouput**: Reconstructed function $\tilde{y}$, represented implicitly
  1: Let $\bar{\mathbf{y}} \in \mathbb{C}^n$ be the vector with $\bar{y}_i = v_i \cdot [y(t_i) + z(t_i)]$
  2: **return** $\tilde{\boldsymbol{\alpha}} := (\boldsymbol{K}_\mu + \varepsilon \boldsymbol{I})^{-1} \bar{\mathbf{y}}$.

---

For any $t$ in $[0, T]$, we can evaluate $\tilde{y}(t)$ by computing $k_\mu(t_i, t)$ for all $i \in 1, \ldots, n$ and returning $\tilde{y}(t) = \sum_{i=1}^n \tilde{\alpha}_i \cdot k_\mu(t_i, t)$.

In order to start off the analysis, we show that solving a Fourier operator analogue to a Ridge Regression problem guarantees a solution to Problem 2. That is, we reduce the problem of finding a good interpolant to the problem of solving a specialized Ridge Regression problem. However, this Ridge Regression problem involves an operator on $[0, T]$, and is not in terms of samples observed. So, we then have to bound how many samples we need to observe for our samples to generalize well to the continuous $[0, T]$ domain, for all PDFs $\mu \in \mathcal{U}$. This is the core technical challenge of this paper.

**Claim 1.** *Let $\tilde{\mu} \in \mathcal{U}$ and $\tilde{g} \in L_2(\mu)$ be near-optimal solutions to a continuous-time Fourier Fitting problem with ridge regularization:*

$$\|\mathcal{F}_{\tilde{\mu}}^* \tilde{g} - (y+z)\|_T^2 + \varepsilon \|\tilde{g}\|_{\tilde{\mu}}^2 \leq C \min_{\mu \in \mathcal{U}} \min_{g \in L_2(\mu)} \left[ \|\mathcal{F}_\mu^* g - (y+z)\|_T^2 + \varepsilon \|g\|_\mu^2 \right]$$

*Let $\hat{\mathcal{U}} \subseteq \mathcal{U}$ be the subset of PDFs that are able of representing $y$ exactly. Then,*

$$\|y - \tilde{y}\|_T^2 \leq 2(C+1)\|z\|_T^2 + 2C\varepsilon \min_{\mu \in \hat{\mathcal{U}}, \, y = \mathcal{F}_\mu^* h} \|h\|_\mu^2$$

This claim is proven in Appendix A, and directly generalizes the proof of Claim 4 in [AKM$^+$19]. Our goal is now to find a $\tilde{\mu}$ and $\tilde{g}$ that approximately minimize $\|\mathcal{F}_\mu^* g - (y+z)\|_T^2 + \varepsilon \|g\|_\mu^2$. If we only had one $\mu$ to consider, the prior work would be able to solve this with $O(s_{\mu,\varepsilon}(\frac{1}{\delta} + \log(s_{\mu,\varepsilon})))$ many samples. However, since our goal is to analyze hyperparameter tuning, we consider the cases with both exponentially large and infinitely large $\mathcal{U}$. In these cases, union bounds using prior work would yield exponentially large and unbounded sample complexities, respectively. In order to avoid this, we form an epsilon-net style argument. The argument follows in two steps:

1. Sampling Time with Finitely Many PDFs: Assume that $\mathcal{U}$ is finite. Let $s_{\max,\varepsilon}$ be the largest statistical dimension found in $\mathcal{U}$. Then we prove that $O(s_{\max,\varepsilon} \log(\frac{s_{\max,\varepsilon}}{\delta} \cdot |\mathcal{U}|))$ observations suffice to recover a near-optimal $(\tilde{\mu}, \tilde{g})$ pair. We emphasize the logarithmic dependence on $|\mathcal{U}|$, since this will allow us to consider exponentially large sets in the next step.

2. Discretization of Kernel Hyperparameters: Assume that $\mathcal{U}$ is the set of Gaussian Mixture PDFs with $q$ Gaussians, taking means in $[-W, W]$, lengthscales in $[m, M]$, and weights in $[0, 1]$. Then we create a *finite* set of Gaussian Mixture PDFs $\tilde{\mathcal{U}}$ such that the best $(\tilde{\mu}, \tilde{g})$ pair on $\tilde{\mathcal{U}}$ is nearly optimal on all of $\mathcal{U}$. In particular, we find $|\tilde{\mathcal{U}}| = O((\frac{W}{m} \log(\frac{M}{m}))^q)$.

Our result from the first bullet point allows us to handle the exponentially large set $\tilde{\mathcal{U}}$ created in the second bullet point. After combining these results and noting that $s_{\max,\varepsilon} = \tilde{O}(qMT)$, we find that $\tilde{O}(q^2 MT \log(\frac{W}{m}))$ time samples suffice to identify a near-optimal SM kernel's hyperparameters. The rest of this paper breaks down and explains these two theoretical results in detail.

## 4 Sampling Time with Finitely Many PDFs

In this section we assume that the given set of PDFs $\mathcal{U}$ is finite, and let $Q := |\mathcal{U}|$. Let $\tilde{y}$ and $\tilde{\mu} \in \mathcal{U}$ be the KRR interpolant and associated PDF that minimize our sample ridge regression cost. We then prove that $\tilde{y}$ describes a nearly-optimal interpolant that satisfies the requirement of Claim 1, so long as we take sufficient samples from the Universal Sampling Distribution (Definition 2). In particular, if $s_{\max,\varepsilon}$ is the largest statistical dimension found in $\mathcal{U}$, then we require $O(s_{\max,\varepsilon} \log(\frac{s_{\max,\varepsilon}}{\delta} \cdot Q))$ samples. We formally state this first core technical result:

**Theorem 1.** *Let $\mathcal{U}$ be a finite set of PDFs. Let $s_{\max,\varepsilon}$ be the maximum statistical dimension in $\mathcal{U}$. Let Algorithm 1 output observation times $t_1, \ldots, t_n$, weights $v_1, \ldots, v_n$, and weighted Kernel Matrices $\boldsymbol{K}_{\mu_1}, \ldots, \boldsymbol{K}_{\mu_Q}$. Let $\bar{\mathbf{y}}$ be the observed response vector. Let $\tilde{\mu}, \tilde{\boldsymbol{\alpha}}$ solve the ridge regression problem:*

$$\tilde{\mu}, \tilde{\boldsymbol{\alpha}} := \underset{\mu \in \mathcal{U}, \boldsymbol{\alpha} \in \mathbb{R}^n}{\operatorname{argmin}} \|\boldsymbol{K}_\mu \boldsymbol{\alpha} - \bar{\mathbf{y}}\|_2^2 + \varepsilon \boldsymbol{\alpha}^\intercal \boldsymbol{K}_\mu \boldsymbol{\alpha} \tag{2}$$

*Define the Fourier domain version of the interpolant[4]: $\tilde{g}(\xi) := \sum_{j=1}^n v_j \tilde{\alpha}_j e^{-2\pi i \xi t_j}$. If $n = \Omega(s_{\max,\varepsilon} \log(\frac{s_{\max,\varepsilon}}{\delta} \cdot Q))$, then with probability $1 - \delta$ we have*

$$\|\mathcal{F}_{\tilde{\mu}}^* \tilde{g} - (y + z)\|_T^2 + \varepsilon \|\tilde{g}\|_{\tilde{\mu}}^2 \leq \left(72 + \tfrac{18}{\delta}\right) \min_{\mu \in \mathcal{U}} \min_{g \in L_2(\mu)} \|\mathcal{F}_\mu^* g - (y + z)\|_T^2 + \varepsilon \|g\|_\mu^2$$

Theorem 1 is proven in Appendix C.2, with a simplified and more approachable proof for the matrix case in Appendix C.1. Intuitively, Theorem 1 states that despite having *adversarial noise*, choosing from a large family of kernels during hyperparameter tuning does not sharply increase the sample complexity of fitting $y(t)$. In other words, Theorem 1 states that $\Omega(s_{\max,\varepsilon} \log(\frac{s_{\max,\varepsilon}}{\delta} \cdot Q))$ samples guarantees a solution to Problem 2 when $\mathcal{U}$ is finite.

In prior work, [AKM$^+$19] proves that $\Omega(s_{\mu,\varepsilon} \log(s_{\mu,\varepsilon} + \frac{1}{\delta}))$ samples guarantees a solution to Problem 1, and that this bound is tight for many common kernels. Since Problem 2 reduces to Problem 1 when $Q = 1$, the sample complexity in Theorem 1 must be tight up to logarithmic factors. Additionally, note that union bounding this result from [AKM$^+$19] over the $Q$ kernels would yield a sample complexity linear in $Q$, instead of the logarithmic rate we prove. This logarithmic rate is important, since the next section will take $Q$ to be exponentially large.

It remains unclear if the dependence on $\frac{1}{\delta}$ in the approximation error is neccessary if we want a logarithmic sample complexity dependence on $Q$. The standard randomized numerical linear algebra approach method requires $O(Q)$ many matrix-multiplication claims, which then incurs a $O(\frac{Q}{\delta})$ sample complexity. This remains as an interesting open problem even in the case of least squares regression, where we choose one of $Q$ different design matrices.

So, Theorem 1 tells us that we can choose from a finite set of kernels without worry, but practitioners do not consider finite sets of kernels in practice. Instead, they usually consider fitting kernels like the SM Kernel, which is parameterized by several *continuous real-valued parameters*. So, we cannot directly apply Theorem 1 to SM Kernel fitting; one more step is needed.

## 5 Discretization of Spectral Mixture Hyperparameters

We now return to the original goal of hyperparameter tuning for kernels. At a high level, we expect that a sufficiently small change to a kernel's hyperparameters should not substantially impact the quality of the kernel as an interpolant. So, instead of considering the continuous range of all hyperparameters, we create a *finite* "net" of hyperparameters $\tilde{\mathcal{U}}$. In particular, any selection of hyperparameters $\hat{\mu} \in \mathcal{U}$ has a corresponding selection of hyperparameters $\tilde{\mu}$ that lies in the net $\tilde{\mathcal{U}}$. Since we design $\tilde{\mu}$ to be sufficiently similar to $\hat{\mu}$, we can then prove that $\hat{\mu}$ cannot achieve a much smaller error than $\tilde{\mu}$. Intuitively, we can think $\tilde{\mathcal{U}}$ as being a discretization of the full continuous set of hyperparameters $\mathcal{U}$.

Then, once we have constructed the discretization $\tilde{\mathcal{U}}$, we can then use Theorem 1 to prove that $n = O(s_{\max,\varepsilon} \log(\frac{s_{\max,\varepsilon}}{\delta} \cdot |\tilde{\mathcal{U}}|))$ observations suffice to interpolate $y$ with a near-optimal choice of hyperparameters. Since Theorem 1 admits a logarithmic dependence on the size of our net $|\tilde{\mathcal{U}}|$, we can create an exponentially large net while achieving polynomial sample complexity bounds.

This broad principle of discretization can easily apply to many kernels; for demonstration purposes, we only consider the SM Kernel in this work. If the reader would like to bound the sample complexity of other kernels, they would only need to form a bound like Theorem 2 below. Here we assume that $\mathcal{U}$ is the set of Gaussian Mixture hyperparameters, mixing $q$ Gaussians with means in $[-W, W]$, lengthscales in $[m, M]$, and weights in $[0, 1]$.

**Theorem 2.** *Fix the constants $W, m, M$ as described above. Define the discretization set for means as*

$$\mathcal{C} := \{-W, -W + m, -W + 2m, \ldots, (k-2)m, W\}$$

*and the discretization set for lengthscales as*

$$\mathcal{S} := \{m, 2m, 4m, 8m, \ldots, 2^{\ell-3}m, M, 2M\}$$

*where $k = \lfloor \frac{2W}{m} \rfloor = |\mathcal{C}|$ and $\ell = \lfloor \log_2(M/m) \rfloor + 1 = |\mathcal{S}|$. Then we have*

$$\min_{\substack{g \in L_2(\mu_{\mathbf{c},\boldsymbol{\sigma}}): \\ \mathbf{c} \in \mathcal{C}^q \\ \boldsymbol{\sigma} \in \mathcal{S}^q}} \|\mathcal{F}_{\mathbf{c},\boldsymbol{\sigma}}^* \tilde{g} - (y+z)\|_T^2 + \varepsilon \|\tilde{g}\|_{\mathbf{c},\boldsymbol{\sigma}}^2 \leq 8 \cdot \min_{\substack{g \in L_2(\mu_{\mathbf{c},\boldsymbol{\sigma},\mathbf{w}}): \\ \mathbf{c} \in [-W,W]^q \\ \boldsymbol{\sigma} \in [m,M]^q \\ \mathbf{w} \in [0,1]^q}} \|\mathcal{F}_{\mathbf{c},\boldsymbol{\sigma},\mathbf{w}}^* \tilde{g} - (y+z)\|_T^2 + \varepsilon \|\tilde{g}\|_{\mathbf{c},\boldsymbol{\sigma},\mathbf{w}}^2$$

Theorem 2 is proven in Appendix B. Note that we take $\mathbf{w}$ to be the all-ones vector without loss of generality. Statistically, increasing the scale of the kernel matrix (i.e. increasing the weight $w_j$) uniquely increases the statistical dimension and decreases the mean squared error. So, as long as we have enough samples to satisfy the statistical dimension requirement when $\mathbf{w}$ is the all ones vector, we should take $\mathbf{w}$ to be all-ones without loss of generality.

In Gaussian Process Regression, the weights in $\mathbf{w}$ are regularized by the likelihood function, which does not apply in the current setting. Instead, since we assume have enough samples to satisfy the all-ones $\mathbf{w}$ vector, we do not need to regularize against the norm of the $\mathbf{w}$ vector.

Intuitively, Theorem 2 reduces the search space for SM kernel hyperparameters down to a finite set of kernels. This allows us to apply Theorem 1 to general SM Kernel fitting. Using Claim 1 as well, we form the following conclusion on the statistical cost of learning SM Kernel hyperparameters:

**Corollary 3.** *Suppose we want to fit a signal using a SM Kernel with $q$ Gaussians whose means lie in $[0, W]$, lengthscales lie in $[m, M]$, and weights lie in $[0, 1]$. Then, with probability 0.99, $n = \tilde{O}(q^2 MT \log(\frac{W}{m}))$ time samples drawn from the Universal Sampling Distribution suffice to have the KRR interpolant $\tilde{y}$ give*

$$\|y - \tilde{y}\|_T^2 \leq C \cdot \left( \|z\|_T^2 + \varepsilon \min_{\mu \in \hat{\mathcal{U}}, y = \mathcal{F}_\mu^* h} \|h\|_\mu^2 \right)$$

*where $\hat{\mathcal{U}}$ is the set of valid SM kernels capable of representing $y$.*

*Proof.* [AKM$^+$19] shows that the statistical dimension of a mixture of $q$ Gaussians is at most $s_{\max,\varepsilon} \leq q \cdot (MT\sqrt{\log(1/\varepsilon)} + \log(1/\varepsilon))$. Theorem 2 tell us that we need to consider $Q = O((\frac{W}{m} \log(\frac{M}{m}))^q)$ specific prior hyperparameters. Then, Theorem 1 tells us that $O(s_{\max,\varepsilon} \log(\frac{s_{\max,\varepsilon}}{\delta} \cdot Q))$ samples suffice to satisfy the precondition for Claim 1, giving us a sample complexity of

$$O\left( q^2 \cdot (MT\sqrt{\log 1/\varepsilon} + \log 1/\varepsilon) \cdot \log\left( \frac{MT\sqrt{\log 1/\varepsilon} + \log 1/\varepsilon}{\delta} \cdot \frac{W}{m} \log\left(\frac{M}{m}\right) \right) \right)$$

$$= \tilde{O}\left( q^2 MT \log\left(\frac{W}{m}\right) \right)$$

$\square$

Note that the $\tilde{O}$ notation hides a logarithmic dependence on $\frac{1}{\varepsilon}$ and a sublogarithmic dependence on $\frac{M}{m}$. Further note that [AKM$^+$19] proves that a single Gaussian kernel with lengthscale $M$ would already require $\tilde{O}(MT)$ samples, so hyperparameter tuning for a single Gaussian only increases the sample complexity by logarithmic factors. However, when we consider multiple Gaussians, our analysis does introduce an extra factor of $q$ beyond statistical dimension $s_{\max,\varepsilon} = \tilde{O}(qMT)$.

## 6 Conclusion

Despite how useful SM Kernels are [WA13, WGNC13, YWSS15, HSSM15, TBT15], practitioners find that tuning SM Kernels is hard in practice [WN15, BMS$^+$19, BHH$^+$16, HDS17, WDLX15]. A practitioner could consider two reasons why it is hard to fit the SM Kernel: either they have too little data to information-theoretically find a good model, or their algorithms fail to find such a model despite having enough information. Our final result, Corollary 3, shows that the statistical complexity

of learning the SM kernel's hyperparameters is not too large, even against adversarial noise. A natural conclusion is that practitioners should likely place effort in finding more effective algorithms.

We see several interesting potential future directions for this work. First, this paper focuses its applications to the Spectral Mixture kernel. Other popular kernels like the Matern, sinc, and Rational Quadratic kernels can also be analyzed under our framework. Further, we provide statistical bounds for finding optimal hyperparameters by designing a discrete optimization problem over exponentially many PDFs, but we do not provide any polynomial time algorithm to solve this problem. We frame hyperparameter optimization as a Fourier fitting problem, so we suspect that such an algorithm can come from recent research in signal processing. In particular, the compressed sensing [BD09, TGS06, BCDH10, CKPS16] and sparse Fourier transform [GIIS14, HIKP12] and literature provides fast algorithms with strong guarantees for many Fourier fitting problems. Lastly, we would like to know if the dependence on $\frac{1}{\delta}$ in the approximation error of Theorem 1 is necessary if we want a logarithmic dependence on $Q$ in the sample complexity.

## Acknowledgments and Disclosure of Funding

We have no funding to disclose.

## Broader Impact

This is a theoretical paper, so discussion of broader impacts has to speculate on future applications for this broad line of research. In particular, this paper considers active regression that is robust and sample-efficient. In other words, we advance our understanding of learning patterns when samples are expensive and noise is large. We speculate on the various impacts of machine learning in these high-sample-cost and high-noise settings.

One of the most common guiding positive benefits is the benefit to medical imaging, where instruments have biased noisy measurements and are expensive to run. So, having ML techniques with low sample complexity and high robustness can give medical practitioners high confidence in their medical conclusions while keeping costs low.

However, this broad framework can just as easily apply to large-scale illegal surveillance through "Internet of Things" (IoT) devices. Many IoT devices are known to have cheap microphones, weak security, and internet access. Well crafted internet crawling code could plausibly access a massive number of such IoT devices, and hence be able to access a massive number of private microphones.

Without progress in our line of research, it may be prohibitively expensive to process or store all the speech heard from all of these microphones. Alternatively, the internet download patterns might be too noticeable for such an operation to secretly run at a large scale. However, progress toward statistically efficient algorithms in the high-sample-cost and high-noise regime might allow code that sends very few messages online while still transmitting all the interesting audio to a third party. This would allow such illegal surveillance on a massive scale.

Both of these settings, the medical and the surveillance, look equivalent from our current level of mathematical/statistical abstraction. Notably, our current research is still too abstract to directly benefit either application. It will require more research to connect results like ours to either the medical or the surveillance applications. Note that the specific examples of medical imaging and IoT surveillance are just two examples of highly positive or highly negative ML applications. So, *under the assumption that the research community will **focus on** positive applications like medical imaging while **avoiding** negative applications like large scale IoT surveillance*, we believe that the benefits of our theoretical research outweigh the negative potential ramifications. Admittedly, this may be an optimistic assumption.

## Footnotes

[1]Throughout this paper, $\mu$ will sometimes denote a scaled PDF that integrates to a constant other than 1.

[2]The polynomial factors on $\alpha$ can be tightened using some recent papers [Erd17, CP19b], but this would only tighten constants in $\int_0^T \tilde{\tau}_\alpha(t)dt$, and hence only tighten constants in the sample complexity.

[3]This is a technical nuance to handle the edge-case that $y(t)$ might not be representable by all of the given PDFs. For instance, if $y$ is a sinusoid with frequency 1, then a bandlimited $\mu$ supported on frequencies 2 through 4 is incapable to of representing $y$ exactly.

[4]This parametrization simply ensures that $\tilde{y}(t) = [\mathcal{F}_{\tilde{\mu}}^* \tilde{g}](t)$

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
