[Supplementary Material]

## A  Interpolation Gaurantees

**Claim 1 Restated.** *Let $\tilde{\mu} \in \mathcal{U}$ and $\tilde{g} \in L_2(\mu)$ be near-optimal solutions to a continuous time Fourier Fitting problem with ridge regression:*

$$\|\mathcal{F}_{\tilde{\mu}}^* \tilde{g} - (y+z)\|_T^2 + \varepsilon \|\tilde{g}\|_{\tilde{\mu}}^2 \le C \min_{\mu \in \mathcal{U}} \min_{g \in L_2(\mu)} \left[ \|\mathcal{F}_\mu^* g - (y+z)\|_T^2 + \varepsilon \|g\|_\mu^2 \right]$$

*Let $\hat{\mathcal{U}} \subseteq \mathcal{U}$ be the subset of PDFs that are able of representing $y$ exactly. Then, letting $\tilde{y} = \mathcal{F}_{\tilde{\mu}}^* \tilde{g}$,*

$$\|y - \tilde{y}\|_T^2 \le 2(C+1)\|z\|_T^2 + 2C\varepsilon \min_{\substack{\mu \in \hat{\mathcal{U}} \\ y = \mathcal{F}_\mu^* h}} \|h\|_\mu^2$$

*Proof.* Letting $y = \mathcal{F}_\mu^* h_\mu$ for all $\mu \in \hat{\mathcal{U}}$, we know that

$$\min_{\mu \in \mathcal{U}} \min_{g \in L_2(\mu)} \left[ \|\mathcal{F}_\mu^* g - (y+z)\|_T^2 + \varepsilon \|g\|_\mu^2 \right] \le \min_{\mu \in \hat{\mathcal{U}}} \left[ \|\mathcal{F}_\mu^* h_\mu - (y+z)\|_T^2 + \varepsilon \|h_\mu\|_\mu^2 \right]$$

$$= \min_{\mu \in \hat{\mathcal{U}}} \left[ \|z\|_T^2 + \varepsilon \|h_\mu\|_\mu^2 \right]$$

$$= \|z\|_T^2 + \varepsilon \min_{\mu \in \hat{\mathcal{U}}} \|h_\mu\|_\mu^2$$

So, using our pair $\tilde{\mu}, \tilde{g}$, we have

$$\|\mathcal{F}_{\tilde{\mu}}^* \tilde{g} - (y+z)\|_T^2 + \varepsilon \|\tilde{g}\|_{\tilde{\mu}}^2 \le C\|z\|_T^2 + C\varepsilon \min_{\mu \in \hat{\mathcal{U}}} \|h_\mu\|_\mu^2$$

Next, by the triangle inequality, and recalling that $\tilde{y} = \mathcal{F}_{\tilde{\mu}}^* \tilde{g}$,

$$\|\tilde{y} - y\|_T - \|z\|_T \le \|\mathcal{F}_{\tilde{\mu}}^* \tilde{g} - (y+z)\|_T$$

$$\|\tilde{y} - y\|_T \le \|z\|_T + \sqrt{C\|z\|_T^2 + C\varepsilon \min_{\mu \in \hat{\mathcal{U}}} \|h_\mu\|_\mu^2}$$

$$\|\tilde{y} - y\|_T^2 \le 2(C+1)\|z\|_T^2 + 2C\varepsilon \min_{\mu \in \hat{\mathcal{U}}} \|h_\mu\|_\mu^2$$

where the last line uses the AM-GM inequality to bound

$$2 \cdot \|z\|_T \cdot \sqrt{C\|z\|_T^2 + C\varepsilon \min_{\mu \in \hat{\mathcal{U}}} \|h_\mu\|_\mu^2} \le \|z\|_T^2 + C\|z\|_T^2 + C\varepsilon \min_{\mu \in \hat{\mathcal{U}}} \|h_\mu\|_\mu^2$$

$\square$

## B  Spectral Mixture Bounds

We are allowed to use $c \in [0, W]$, $\sigma \in [m, M]$, and $w \in [0, 1]$, where $0 < m < M$, and want to have at most $O(1)$ error from our discretization. We start by showing that without loss of generality, we should always take $w$ to be the all-ones vector.

**Lemma 1.** *Let $\mu_1$ and $\mu_2$ be associated with kernel operators $\mathcal{K}_{\mu_1}$ and $\mathcal{K}_{\mu_2}$ such that $\mathcal{K}_{\mu_1} \preceq \mathcal{K}_{\mu_2}$. Then,*

$$\min_{g \in L_2(\mu_1)} \|\mathcal{F}_{\mu_1}^* g - \bar{y}\|_T^2 + \varepsilon \|g\|_{\mu_1}^2 \le \min_{g \in L_2(\mu_2)} \|\mathcal{F}_{\mu_2}^* g - \bar{y}\|_T^2 + \varepsilon \|g\|_{\mu_2}^2$$

*Proof.* For now, consider a arbitrary $\mu$. Note from Lemma 38 of [AKM+19], we know that the minimizer of

$$\min_{g \in L_2(\mu)} \|\mathcal{F}_\mu^* g - y\|_T^2 + \varepsilon \|g\|_\mu^2$$

has the form $\hat{g} = \mathcal{F}_\mu (\mathcal{K}_\mu + \varepsilon \mathcal{I}_T)^{-1} \bar{y}$. Then, we can write

$$\mathcal{F}_\mu^* \hat{g} = \mathcal{K}_\mu (\mathcal{K}_\mu + \varepsilon \mathcal{I}_T)^{-1} \bar{y}$$

$$\|\mathcal{F}_\mu^* - \bar{y}\|_T^2 = \langle \mathcal{K}_\mu (\mathcal{K}_\mu + \varepsilon \mathcal{I}_T)^{-1} \bar{y} - \bar{y}, \mathcal{K}_\mu (\mathcal{K}_\mu + \varepsilon \mathcal{I}_T)^{-1} \bar{y} - \bar{y} \rangle_T$$

$$= \|\bar{y}\|_T^2 - 2\langle \bar{y}, \mathcal{K}_\mu (\mathcal{K}_\mu + \varepsilon \mathcal{I}_T)^{-1} \bar{y} \rangle_T + \langle \mathcal{K}_\mu (\mathcal{K}_\mu + \varepsilon \mathcal{I}_T)^{-1} \bar{y}, \mathcal{K}_\mu (\mathcal{K}_\mu + \varepsilon \mathcal{I}_T)^{-1} \bar{y} \rangle_T$$

$$\|\hat{g}\|_\mu^2 = \langle \mathcal{K}_\mu (\mathcal{K}_\mu + \varepsilon \mathcal{I}_T)^{-1} \bar{y}, \mathcal{K}_\mu (\mathcal{K}_\mu + \varepsilon \mathcal{I}_T)^{-1} \bar{y} \rangle_T$$

Noting that all the inner products on the last two lines share the same right hand side, we find that the value of the true minimizer is

$$
\begin{aligned}
\|\mathcal{F}_\mu^* - \bar{y}\|_T^2 + \varepsilon\|\hat{g}\|_\mu^2 &= \|\bar{y}\|_T^2 + \langle -2\bar{y} + \mathcal{K}_\mu(\mathcal{K}_\mu + \varepsilon\mathcal{I}_T)^{-1}\bar{y} + \varepsilon(\mathcal{K} + \varepsilon\mathcal{I}_T)^{-1}\bar{y}, \mathcal{K}_\mu(\mathcal{K}_\mu + \varepsilon\mathcal{I}_T)^{-1}\bar{y}\rangle_T \\
&= \|\bar{y}\|_T^2 + \langle ((-2(\mathcal{K}_\mu + \varepsilon\mathcal{I}_T) + \mathcal{K}_\mu + \varepsilon\mathcal{I}_T)(\mathcal{K}_\mu + \varepsilon\mathcal{I}_T)^{-1}\bar{y}, \mathcal{K}_\mu(\mathcal{K}_\mu + \varepsilon\mathcal{I}_T)^{-1}\bar{y}\rangle_T \\
&= \|\bar{y}\|_T^2 + \langle -1 \cdot (\mathcal{K}_\mu + \varepsilon\mathcal{I}_T) \cdot (\mathcal{K}_\mu + \varepsilon\mathcal{I}_T)^{-1}\bar{y}, \mathcal{K}_\mu(\mathcal{K}_\mu + \varepsilon\mathcal{I}_T)^{-1}\bar{y}\rangle_T \\
&= \|\bar{y}\|_T^2 - \langle \bar{y}, \mathcal{K}_\mu(\mathcal{K}_\mu + \varepsilon\mathcal{I}_T)^{-1}\bar{y}\rangle_T \\
&= \langle \bar{y}, \mathcal{I}_T - \mathcal{K}_\mu(\mathcal{K}_\mu + \varepsilon\mathcal{I}_T)^{-1}\bar{y}\rangle_T
\end{aligned}
\tag{3}
$$

Then, since we know that the kernel operator $\mathcal{K}_\mu \succeq 0$, we conclude that $\mathcal{I}_T - \mathcal{K}_\mu(\mathcal{K}_\mu + \varepsilon\mathcal{I}_T)^{-1} \succeq 0$ for all kernel operators $\mathcal{K}_\mu$. Additionally, note that Equation 3 is in the analogous form to $\mathbf{x}^\mathsf{T} A\mathbf{x}$ for matrices. In particular, if we decrease the semidefinite order of $\mathcal{I}_T - \mathcal{K}_\mu(\mathcal{K}_\mu + \varepsilon\mathcal{I}_T)^{-1}$, then we decrease the overall minimum value for all signals $\bar{y}$. Since $\mathcal{K}_{\mu_1} \preceq \mathcal{K}_{\mu_2}$, we know that $\mathcal{K}_{\mu_1}(\mathcal{K}_{\mu_1} + \varepsilon\mathcal{I}_T)^{-1} \preceq \mathcal{K}_{\mu_2}(\mathcal{K}_{\mu_2} + \varepsilon\mathcal{I}_T)^{-1}$, and hence

$$
\langle \bar{y}, \mathcal{I}_T - \mathcal{K}_{\mu_2}(\mathcal{K}_{\mu_2} + \varepsilon\mathcal{I}_T)^{-1}\bar{y}\rangle_T \le \langle \bar{y}, \mathcal{I}_T - \mathcal{K}_{\mu_1}(\mathcal{K}_{\mu_1} + \varepsilon\mathcal{I}_T)^{-1}\bar{y}\rangle_T
$$

Or, equivalently,

$$
\min_{g \in L_2(\mu_1)} \|\mathcal{F}_{\mu_1}^* g - \bar{y}\|_T^2 + \varepsilon\|g\|_{\mu_1}^2 \le \min_{g \in L_2(\mu_2)} \|\mathcal{F}_{\mu_2}^* g - \bar{y}\|_T^2 + \varepsilon\|g\|_{\mu_2}^2
$$

$\square$

We now show why this tell us to pick the all-ones vector for SM Kernels:

**Corollary 4.** *Let $\mu_{\mathbf{c},\boldsymbol{\sigma},\mathbf{w}}(\xi)$ be a spectral mixture PDF with weights $\mathbf{w} \in [0,1]^q$. Then, the spectral mixture with the same means and lengthscales $\mu_{\mathbf{c},\boldsymbol{\sigma}}(\xi)$ achieves uniquely less error:*

$$
\min_{g \in L_2(\mu_{\mathbf{c},\boldsymbol{\sigma},\mathbf{w}})} \|\mathcal{F}_{\mathbf{c},\boldsymbol{\sigma},\mathbf{w}}^* g - \bar{y}\|_T^2 + \varepsilon\|g\|_{\mathbf{c},\boldsymbol{\sigma},\mathbf{w}}^2 \le \min_{g \in L_2(\mu_{\mathbf{c},\boldsymbol{\sigma}})} \|\mathcal{F}_{\mathbf{c},\boldsymbol{\sigma}}^* g - \bar{y}\|_T^2 + \varepsilon\|g\|_{\mathbf{c},\boldsymbol{\sigma}}^2
$$

*Proof.* Note that the Kernel operator associated with $\mu_{\mathbf{c},\boldsymbol{\sigma}\mathbf{w}}$ is $\mathcal{K}_{\mathbf{c},\boldsymbol{\sigma},\mathbf{w}} = \sum_{j=1}^q w_j \mathcal{K}_{c_j,\sigma_j}$. Since $w_j \le 1$, we find that $\mathcal{K}_{\mathbf{c},\boldsymbol{\sigma},\mathbf{w}} \preceq \sum_{j=1}^q \mathcal{K}_{c_j,\sigma_j} = \mathcal{K}_{\mathbf{c},\boldsymbol{\sigma}}$, the kernel operator associated with the all ones weight vector. So, by Lemma 1, we complete the proof. $\square$

With this reduction in place, we move onto consider the means and lengthscales of our kernel. We are allowed to use means $c \in [0, W]$ and lengthscales $\sigma \in [m, M]$, where $0 < m < M$, and want to have at most $O(1)$ error from our discretization. We achieve this with additive mean step sizes and multiplicative lengthscale step sizes,:

$$
\begin{aligned}
\mathcal{C} &= \{0, \rho m, 2\rho m, \dots, (k-2)\rho m, W\} \\
\mathcal{S} &= \{m, (1+\gamma)m, (1+\gamma)^2 m, \dots, (1+\gamma)^{\ell-3} m, M, (1+\gamma)M\}
\end{aligned}
$$

Note that the step sizes for both the means and lengthscales are left in terms of the minimum lengthscale. The set $\mathcal{C}$ guarantees that any $\hat{c} \in [0, W]$ has $\tilde{c} \in \mathcal{C}$ such that

$$
|\tilde{c} - \hat{c}| \le \rho m
\tag{4}
$$

Additionally, set $\mathcal{S}$ guarantees that any $\hat{\sigma} \in [m, M]$ has $\tilde{\sigma} \in \mathcal{S}$ such that

$$
\frac{\tilde{\sigma}}{(1+\gamma)^2} \le \hat{\sigma} \le \frac{\tilde{\sigma}}{1+\gamma} < \tilde{\sigma}
\tag{5}
$$

Notably, we do not allow $\hat{\sigma}$ to be arbitrarily close to $\tilde{\sigma}$, but instead guarantee a multiplicative gap between the two. This is why the maximum value of $\mathcal{S}$ is greater than $M$. There are $k = \left\lfloor \frac{W}{\rho m} \right\rfloor$ means in $\mathcal{C}$ and $\ell = \left\lfloor \frac{\ln(2M/m)}{\ln(1+\gamma)} \right\rfloor$ lengthscales in $\mathcal{S}$. We now discretize a SM kernel of $q$ Gaussian modes by rounding to means in $\mathcal{C}$ and lengthscales in $\mathcal{S}$:

**Lemma 2.**

$$\min_{\substack{\mathbf{c}\in\mathcal{C}^q \\ \boldsymbol{\sigma}\in\mathcal{S}^q}} \min_{g\in L_2(\mu_{\mathbf{c},\boldsymbol{\sigma}})} \|\mathcal{F}_{\mathbf{c},\boldsymbol{\sigma}}^* g - \bar{y}\|_T^2 + \varepsilon\|g\|_{\mathbf{c},\boldsymbol{\sigma}}^2 \le C \min_{\substack{\mathbf{c}\in[0,W]^q \\ \boldsymbol{\sigma}\in[m,M]^q}} \min_{g\in L_2(\mu_{\mathbf{c},\boldsymbol{\sigma}})} \|\mathcal{F}_{\mathbf{c},\boldsymbol{\sigma}}^* g - \bar{y}\|_T^2 + \varepsilon\|g\|_{\mathbf{c},\boldsymbol{\sigma}}^2$$

$$(6)$$

*Where* $C = (1+\gamma)^2 \exp(\frac{\rho^2}{2} \cdot \frac{1}{1-\frac{1}{(1+\gamma)^2}})$.

If we want a factor of 3 error, we can take $\gamma = \rho = 0.5$, so that $C \approx 2.8178 < 3$. This makes $|\mathcal{C}| = O(\frac{W}{m})$ and $|\mathcal{S}| = O(\log(M/m))$, so that the discretize space of $q$ SM kernels has $O((\frac{W}{m}\log(\frac{M}{m}))^q)$ choices of hyperparameter to consider.

*Proof.* Let $\hat{\mathbf{c}}, \hat{\boldsymbol{\sigma}}$, and $\hat{g}$ be the minimizers of the right hand side of Inequality 6. Let $p(\xi; c, \sigma) := \frac{1}{\sqrt{2\pi\sigma^2}}e^{-\frac{(\xi-c)^2}{2\sigma^2}}$ be the Gaussian PDF with mean $c$ and lengthscale $\sigma^2$. Further, let $p(\xi; \mathbf{c}, \boldsymbol{\sigma}) := \sum_{j=1}^q p(\xi, c_j, \sigma_j)$ be the sum of the Gaussians described in $\mathbf{c}$ and $\boldsymbol{\sigma}$. This allows us to write $d\mu_{\mathbf{c},\boldsymbol{\sigma}}(\xi) = p(\xi; \mathbf{c}, \boldsymbol{\sigma})d\xi$.

Let $\tilde{\mathbf{c}}$ and $\tilde{\boldsymbol{\sigma}}$ be the discretizations of $\hat{\mathbf{c}}$ and $\hat{\boldsymbol{\sigma}}$ using the schemes from Equation 4 and Equation 5. Let $\tilde{g}$ be the following rounding of $\hat{g}$:

$$\tilde{g}(\xi) := \hat{g}(\xi) \cdot \frac{p(\xi; \hat{\mathbf{c}}, \hat{\boldsymbol{\sigma}})}{p(\xi; \tilde{\mathbf{c}}, \tilde{\boldsymbol{\sigma}})}$$

Then, this particular rounding implies that the Inverse Fourier Transform of $\tilde{g}$ preserves the Inverse Fourier Transform of $\hat{g}$:

$$
\begin{aligned}
[\mathcal{F}_{\tilde{\mathbf{c}},\tilde{\boldsymbol{\sigma}}}^* \tilde{g}](t) &= \int_{\mathbb{R}} \tilde{g}(\xi)e^{2\pi i\xi t}d\mu_{\tilde{\mathbf{c}},\tilde{\boldsymbol{\sigma}}}(\xi) \\
&= \int_{\mathbb{R}} \hat{g}(\xi) \cdot \frac{p(\xi; \hat{\mathbf{c}}, \hat{\boldsymbol{\sigma}})}{p(\xi; \tilde{\mathbf{c}}, \tilde{\boldsymbol{\sigma}})} \cdot e^{2\pi i\xi t} \cdot p(\xi, \tilde{\mathbf{c}}, \tilde{\boldsymbol{\sigma}}) \cdot d\xi \\
&= \int_{\mathbb{R}} \hat{g}(\xi) \cdot p(\xi; \hat{\mathbf{c}}, \hat{\boldsymbol{\sigma}}) \cdot e^{2\pi i\xi t} \cdot d\xi \\
&= \int_{\mathbb{R}} \hat{g}(\xi)e^{2\pi i\xi t}d\mu_{\hat{\mathbf{c}},\hat{\boldsymbol{\sigma}}}(\xi) \\
&= [\mathcal{F}_{\hat{\mathbf{c}},\hat{\boldsymbol{\sigma}}}^* \hat{g}](t)
\end{aligned}
$$

So, we immediately know that $\|\mathcal{F}_{\tilde{\mathbf{c}},\tilde{\boldsymbol{\sigma}}}^* \tilde{g} - \bar{y}\|_T^2 = \|\mathcal{F}_{\hat{\mathbf{c}},\hat{\boldsymbol{\sigma}}}^* \hat{g} - \bar{y}\|_T^2$. All we need to do now is bound the power of $\tilde{g}$ with respect to $\tilde{\mathbf{c}}$ and $\tilde{\boldsymbol{\sigma}}$:

$$
\begin{aligned}
\|\tilde{g}\|_{\tilde{\mathbf{c}},\tilde{\boldsymbol{\sigma}}}^2 &= \int_{\mathbb{R}} |\tilde{g}(\xi)|^2 d\mu_{\tilde{\mathbf{c}},\tilde{\boldsymbol{\sigma}}}(\xi) \\
&= \int_{\mathbb{R}} |\hat{g}(\xi)|^2 \left(\frac{p(\xi; \hat{\mathbf{c}}, \hat{\boldsymbol{\sigma}})}{p(\xi; \tilde{\mathbf{c}}, \tilde{\boldsymbol{\sigma}})}\right)^2 p(\xi; \tilde{\mathbf{c}}, \tilde{\boldsymbol{\sigma}}) \, d\xi \\
&= \int_{\mathbb{R}} |\hat{g}(\xi)|^2 \frac{p(\xi; \hat{\mathbf{c}}, \hat{\boldsymbol{\sigma}})}{p(\xi; \tilde{\mathbf{c}}, \tilde{\boldsymbol{\sigma}})} p(\xi; \hat{\mathbf{c}}, \hat{\boldsymbol{\sigma}}) \, d\xi \\
&\le C \int_{\mathbb{R}} |\hat{g}(\xi)|^2 p(\xi; \hat{\mathbf{c}}, \hat{\boldsymbol{\sigma}}) \, d\xi \\
&= C\|\hat{g}\|_{\hat{\mathbf{c}},\hat{\boldsymbol{\sigma}}}^2
\end{aligned}
$$

The inequality uses the fact that $\frac{p(\xi;\mathbf{c},\hat{\boldsymbol{\sigma}})}{p(\xi;\mathbf{c},\tilde{\boldsymbol{\sigma}})} \le C$ for all $\xi$, proven below.

First, we bound the ratio $\frac{p(\xi;\hat{c}_1,\hat{\sigma}_1)}{p(\xi;\tilde{c}_1,\tilde{\sigma}_1)}$. Note that

$$\frac{p(\xi;\hat{c}_1,\hat{\sigma}_1)}{p(\xi;\tilde{c}_1,\tilde{\sigma}_1)} = \frac{\tilde{\sigma}_1}{\hat{\sigma}_1} \exp\left(\frac{(\xi-\tilde{c}_1)^2}{2\tilde{\sigma}_1^2} - \frac{(\xi-\hat{c}_1)^2}{2\hat{\sigma}_1^2}\right)$$

With some calculus, we can show that the maximum of the right hand side occurs when $\xi = \frac{\hat{\sigma}_1^2 \tilde{c}_1 - \tilde{\sigma}_1^2 \hat{c}_1}{\hat{\sigma}_1^2 - \tilde{\sigma}_1^2}$ and attains hence maximum value

$$\frac{p(\xi; \hat{c}_1, \hat{\sigma}_1)}{p(\xi; \tilde{c}_1, \tilde{\sigma}_1)} \leq \frac{\tilde{\sigma}_1}{\hat{\sigma}_1} \exp\left(\frac{1}{2} \cdot \frac{(\tilde{c}_1 - \hat{c}_1)^2}{\tilde{\sigma}_1^2 - \hat{\sigma}_1^2}\right)$$

We can then use our rounding schemes from Equation 4 and Equation 5 to say

- $\frac{\tilde{\sigma}_1}{\hat{\sigma}_1} \leq \frac{\tilde{\sigma}_1}{\frac{\tilde{\sigma}_1}{(1+\gamma)^2}} = (1+\gamma)^2$

- $\tilde{\sigma}_1^2 - \hat{\sigma}_1^2 \geq \tilde{\sigma}_1^2 - \frac{\tilde{\sigma}_1^2}{(1+\gamma)^2} = \tilde{\sigma}_1^2 (1 - \frac{1}{(1+\gamma)^2}) \geq m^2 (1 - \frac{1}{(1+\gamma)^2})$

- $(\tilde{c}_1 - \hat{c}_1)^2 \leq \rho^2 m^2$

With these three bounds, we conclude

$$\frac{p(\xi; \hat{c}_1, \hat{\sigma}_1)}{p(\xi; \tilde{c}_1, \tilde{\sigma}_1)} \leq (1+\gamma)^2 \exp(\frac{\rho^2}{2} \cdot \frac{1}{1 - \frac{1}{(1+\gamma)^2}}) = C$$

Finally, we complete the proof by noting

$$\begin{aligned}
\frac{p(\xi; \hat{\mathbf{c}}, \hat{\boldsymbol{\sigma}})}{p(\xi; \hat{\mathbf{c}}, \tilde{\boldsymbol{\sigma}})} &= \frac{\sum_{j=1}^n p(\xi; \hat{c}_j, \hat{\sigma}_j)}{p(\xi; \tilde{\mathbf{c}}, \tilde{\boldsymbol{\sigma}})} \\
&\leq \frac{\sum_{j=1}^n C \cdot p(\xi; \tilde{c}_j, \tilde{\sigma}_j)}{p(\xi; \tilde{\mathbf{c}}, \tilde{\boldsymbol{\sigma}})} \\
&= C \cdot \frac{\sum_{j=1}^n p(\xi; \tilde{c}_j, \tilde{\sigma}_j)}{p(\xi; \tilde{\mathbf{c}}, \tilde{\boldsymbol{\sigma}})} \\
&= C \cdot \frac{p(\xi; \tilde{\mathbf{c}}, \tilde{\boldsymbol{\sigma}})}{p(\xi; \tilde{\mathbf{c}}, \tilde{\boldsymbol{\sigma}})} \\
&= C
\end{aligned}$$

$\square$

## C    Multiple Prior Subsampling Bounds

### C.1    Proof for the Matrix Case

First, we introduce the matrix version of the ridge leverage function, first introduced in [AM15]:

**Definition 3.** *For a matrix $\boldsymbol{A} \in \mathbb{R}^{n \times d}$, we define the $\varepsilon$-ridge leverage score for row $i$ as*

$$\tau_{i,\varepsilon}(\boldsymbol{A}) := \max_{\{\boldsymbol{\alpha} \in \mathbb{R}^d : \|\boldsymbol{\alpha}\|_2 > 0\}} \frac{|[\boldsymbol{A}\boldsymbol{\alpha}]_i|^2}{\|\boldsymbol{A}\boldsymbol{\alpha}\|_2^2 + \varepsilon \|\boldsymbol{\alpha}\|_2^2}$$

We first import a result from [CMM17] that shows how ridge leverage score sampling spectrally embeds matrices:

**Imported Theorem 1** (Theorem 5 from [CMM17])**.** *Let $\boldsymbol{A} \in \mathbb{R}^{n \times d}$ and $\varepsilon \geq 0$. Let rows $r_1, \ldots, r_m$ be sampled iid proportionally to $\tilde{\tau}_\varepsilon(i)$, where $\tilde{\tau}_\varepsilon(i) \geq \tau_{i,\varepsilon}(\boldsymbol{A})$. Define $\tilde{s} := \sum_{i=1}^m \tilde{\tau}_\varepsilon(i)$. Let $\boldsymbol{S} \in \mathbb{R}^{m \times n}$ be the sample and rescale matrix: $[\boldsymbol{S}]_{i,j} = \sqrt{\frac{s}{m \tilde{\tau}_\varepsilon(i)}} \cdot \mathbb{1}_{[r_i = j]}$. Then if $m = O(\frac{s \log(s/\delta)}{\Delta^2})$, with probability $1 - \delta$ we have*

$$(1 - \Delta)(\boldsymbol{A}^\mathsf{T}\boldsymbol{A} - \varepsilon \boldsymbol{I}) \preceq \boldsymbol{A}^\mathsf{T}\boldsymbol{S}^\mathsf{T}\boldsymbol{S}\boldsymbol{A} + \varepsilon \boldsymbol{I} \preceq (1 + \Delta)(\boldsymbol{A}^\mathsf{T}\boldsymbol{A} + \varepsilon \boldsymbol{I})$$

Then we move onto the theorem we want to prove:

**Theorem 5.** *Let $\boldsymbol{A}_1, \ldots, \boldsymbol{A}_Q \in \mathbb{R}^{m \times d}$ and $\mathbf{b} \in \mathbb{R}^d$. Fix ridge parameter $\varepsilon \geq 0$. Sample rows $r_1, \ldots, r_n \propto \tilde{\tau}_\varepsilon$ where $\tilde{\tau}_\varepsilon$ is an upper bound for the $\varepsilon$-ridge leverage scores of all pairs of design matrices conjoined: $\tilde{\tau}_\varepsilon(i) \geq \tau_{i,\varepsilon}([\boldsymbol{A}_j, \boldsymbol{A}_k])$ for all $j, k$. Let $\tilde{s}_\varepsilon = \sum_{i=1}^m \tilde{\tau}_\varepsilon(i)$. Build a sample-and-rescale matrix $\boldsymbol{S} \in \mathbb{R}^{n \times m} : \boldsymbol{S}_{j,k} = \sqrt{\frac{\tilde{s}_\varepsilon}{n \tilde{\tau}_\varepsilon(j)}} \mathbb{1}_{[r_j = k]}$. Then let $\tilde{k}, \tilde{\mathbf{x}}$ solve the subsampled regression problem:*

$$\tilde{k}, \tilde{\mathbf{x}} := \underset{k \in [Q], \mathbf{x} \in \mathbb{R}^d}{\operatorname{argmin}} \|\boldsymbol{S}\boldsymbol{A}_k\mathbf{x} - \boldsymbol{S}\mathbf{b}\|_2^2 + \varepsilon\|\mathbf{x}\|_2^2$$

*If $n = O(\tilde{s}_\varepsilon \log(\frac{\tilde{s}_\varepsilon}{\delta} \cdot Q))$, then with probability $1 - \delta$ we have*

$$\|\boldsymbol{A}_{\tilde{k}}\tilde{\mathbf{x}} - \mathbf{b}\|_2^2 + \varepsilon\|\tilde{\mathbf{x}}\|_2^2 \leq (72 + {}^{18}/\delta) \min_{k \in [Q]} \min_{\mathbf{x} \in \mathbb{R}^d} \|\boldsymbol{A}_k\mathbf{x} - \mathbf{b}\| + \varepsilon\|\mathbf{x}\|_2^2$$

The proof of Theorem 1 closely mirrors that of Theorem 5, except that Theorem 1 additionally bounds the Fourier version of the pairwise leverage scores $\tau_{i,\varepsilon}([\boldsymbol{A}_j, \boldsymbol{A}_k])$ and proves a new operator spectral embedding guarantee to handle this case.

*Proof.* First, we define the norm $\|(\mathbf{y}; \mathbf{x})\|_{2,\varepsilon}^2 := \|\mathbf{y}\|_2^2 + \varepsilon\|\mathbf{x}\|_2^2$. Then let $\hat{k}$ and $\hat{\mathbf{x}}$ be the true minimizers for the full optimization problem:

$$\hat{k}, \hat{\mathbf{x}} := \underset{k \in [Q], \mathbf{x} \in \mathbb{R}^d}{\operatorname{argmin}} \|(\boldsymbol{A}_k\mathbf{x} - \mathbf{b}; \mathbf{x})\|_{2,\varepsilon}^2$$

By the triangle inequality, and the inverse triangle inequality, we have for any $\boldsymbol{A}_k$ and any $\mathbf{x}$,

$$\|(\boldsymbol{S}(\boldsymbol{A}_k\mathbf{x} - \mathbf{b}); \mathbf{x})\|_{2,\varepsilon} \in \|(\boldsymbol{S}(\boldsymbol{A}_{\hat{k}}\hat{\mathbf{x}} - \boldsymbol{A}_k\mathbf{x}); \hat{\mathbf{x}} - \mathbf{x})\|_{2,\varepsilon} \pm \|(\boldsymbol{S}(\boldsymbol{A}_{\hat{k}}\hat{\mathbf{x}} - \mathbf{b}); \hat{\mathbf{x}})\|_{2,\varepsilon}$$

We bound these two terms separately, starting with the latter. Letting $\hat{\mathbf{b}}_\perp := \boldsymbol{A}_{\hat{k}}\hat{\mathbf{x}} - \mathbf{b}$, we have

$$\mathbb{E}[\|\boldsymbol{S}(\boldsymbol{A}_{\hat{k}}\hat{\mathbf{x}} - \mathbf{b})\|_2^2] = \mathbb{E}[\|\boldsymbol{S}\hat{\mathbf{b}}_\perp\|_2^2] = \mathbb{E}[\hat{\mathbf{b}}_\perp^\mathsf{T}\boldsymbol{S}^\mathsf{T}\boldsymbol{S}\hat{\mathbf{b}}_\perp] = \hat{\mathbf{b}}_\perp^\mathsf{T}\mathbb{E}[\boldsymbol{S}^\mathsf{T}\boldsymbol{S}]\hat{\mathbf{b}}_\perp$$

And since $\mathbb{E}[\boldsymbol{S}^\mathsf{T}\boldsymbol{S}] = \boldsymbol{I}$, we find $\mathbb{E}[\|(\boldsymbol{S}(\boldsymbol{A}_{\hat{k}}\hat{\mathbf{x}} - \mathbf{b}); \hat{\mathbf{x}})\|_{2,\varepsilon}^2] = \|\hat{\mathbf{b}}_\perp\|_2^2 + \varepsilon\|\hat{\mathbf{x}}\|_2^2 = \|(\boldsymbol{A}_{\hat{k}}\hat{\mathbf{x}} - \mathbf{b}; \hat{\mathbf{x}})\|_{2,\varepsilon}^2$. Hence, by Markov's inequality, we have

$$\|(\boldsymbol{S}(\boldsymbol{A}_{\hat{k}}\hat{\mathbf{x}} - \mathbf{b}); \hat{\mathbf{x}})\|_{2,\varepsilon} \leq \sqrt{\frac{2}{\delta}} \|(\boldsymbol{A}_{\hat{k}}\hat{\mathbf{x}} - \mathbf{b}; \hat{\mathbf{x}})\|_{2,\varepsilon}$$

with probability $1 - \frac{\delta}{2}$.

Next, note that for constant $\Delta$ and for any $\boldsymbol{A}_j, \boldsymbol{A}_k$ we have

$$\|(\boldsymbol{S}[\boldsymbol{A}_j \ \boldsymbol{A}_k]\mathbf{v}; \mathbf{v})\|_{2,\varepsilon} \in (1 \pm \Delta) \|([\boldsymbol{A}_j \ \boldsymbol{A}_k]\mathbf{v}; \mathbf{v})\|_{2,\varepsilon} \text{ for all } \mathbf{v} \in \mathbb{R}^{2d}$$

with probability $1 - \frac{\delta}{2}$. This follows directly from the definition of the ridge norm and the fact that $\boldsymbol{S}$ is generated using upper bounds for leverage scores for $[\boldsymbol{A}_j \ \boldsymbol{A}_k]$, following Imported Theorem 1. Then, we find

$$\begin{aligned}
\left\|\left(\boldsymbol{S}(\boldsymbol{A}_{\hat{k}}\hat{\mathbf{x}} - \boldsymbol{A}_k\mathbf{x}) ; \hat{\mathbf{x}} - \mathbf{x}\right)\right\|_{2,\varepsilon} &= \left\|\left(\boldsymbol{S}\begin{bmatrix}\boldsymbol{A}_{\hat{k}} & \boldsymbol{A}_k\end{bmatrix}\begin{bmatrix}\hat{\mathbf{x}} \\ -\mathbf{x}\end{bmatrix} ; \begin{bmatrix}\hat{\mathbf{x}} \\ -\mathbf{x}\end{bmatrix}\right)\right\|_{2,\varepsilon} \\
&\in (1 \pm \Delta)\left\|\left(\begin{bmatrix}\boldsymbol{A}_{\hat{k}} & \boldsymbol{A}_k\end{bmatrix}\begin{bmatrix}\hat{\mathbf{x}} \\ -\mathbf{x}\end{bmatrix} ; \begin{bmatrix}\hat{\mathbf{x}} \\ -\mathbf{x}\end{bmatrix}\right)\right\|_{2,\varepsilon} \\
&= (1 \pm \Delta)\left\|\left(\boldsymbol{A}_{\hat{k}}\hat{\mathbf{x}} - \boldsymbol{A}_k\mathbf{x} ; \hat{\mathbf{x}} - \mathbf{x}\right)\right\|_{2,\varepsilon}
\end{aligned}$$

Further, by the triangle and inverse triangle inequalities, we have

$$\|(\boldsymbol{A}_{\hat{k}}\hat{\mathbf{x}} - \boldsymbol{A}_k\mathbf{x} ; \hat{\mathbf{x}} - \mathbf{x})\|_{2,\varepsilon} \in \|(\boldsymbol{A}_k\mathbf{x} - \mathbf{b} ; \mathbf{x})\|_{2,\varepsilon} \pm \|(\boldsymbol{A}_{\hat{k}}\hat{\mathbf{x}} - \mathbf{b} ; \hat{\mathbf{x}})\|_{2,\varepsilon}$$

Putting these last two inequalities together, we find that all $\boldsymbol{A}_k$ and $\mathbf{x}$ have

$$\left\|\left(\boldsymbol{S}(\boldsymbol{A}_{\hat{k}}\hat{\mathbf{x}} - \boldsymbol{A}_k\mathbf{x}) ; \hat{\mathbf{x}} - \mathbf{x}\right)\right\|_{2,\varepsilon} \in (1 \pm \Delta)\left\|\left(\boldsymbol{A}_k\mathbf{x} - \mathbf{b} ; \mathbf{x}\right)\right\|_{2,\varepsilon} \pm (1 + \Delta)\left\|\left(\boldsymbol{A}_{\hat{k}}\hat{\mathbf{x}} - \mathbf{b} ; \hat{\mathbf{x}}\right)\right\|_{2,\varepsilon}$$

Then, using this bound, alongside the Markov bound and the original triangle inequality, we find

$$
\begin{aligned}
\left\| \left( \boldsymbol{S}(\boldsymbol{A}_k \mathbf{x} - \mathbf{b}) \,;\, \mathbf{x} \right) \right\|_{2,\varepsilon} &\in \left\| \left( \boldsymbol{S}(\boldsymbol{A}_{\hat{k}}\hat{\mathbf{x}} - \boldsymbol{A}_k \mathbf{x}) \,;\, \hat{\mathbf{x}} - \mathbf{x} \right) \right\|_{2,\varepsilon} \pm \left\| \left( \boldsymbol{S}(\boldsymbol{A}_{\hat{k}}\hat{\mathbf{x}} - \mathbf{b}) \,;\, \hat{\mathbf{x}} \right) \right\|_{2,\varepsilon} \\
&= \left\| \left( \boldsymbol{S}(\boldsymbol{A}_{\hat{k}}\hat{\mathbf{x}} - \boldsymbol{A}_k \mathbf{x}) \,;\, \hat{\mathbf{x}} - \mathbf{x} \right) \right\|_{2,\varepsilon} \pm \sqrt{\tfrac{2}{\delta}} \left\| \left( \boldsymbol{A}_{\hat{k}}\hat{\mathbf{x}} - \mathbf{b} \,;\, \hat{\mathbf{x}} \right) \right\|_{2,\varepsilon} \\
&\subseteq \left( (1 \pm \Delta) \left\| \left( \boldsymbol{A}_{\hat{k}}\hat{\mathbf{x}} - \mathbf{b} \,;\, \hat{\mathbf{x}} \right) \right\|_{2,\varepsilon} \pm (1 + \Delta) \left\| \left( \boldsymbol{A}_k \mathbf{x} - \mathbf{b} \,;\, \mathbf{x} \right) \right\|_{2,\varepsilon} \right) \pm \sqrt{\tfrac{2}{\delta}} \left\| \left( \boldsymbol{A}_{\hat{k}}\hat{\mathbf{x}} - \mathbf{b} \,;\, \hat{\mathbf{x}} \right) \right\|_{2,\varepsilon} \\
&= (1 \pm \Delta) \left\| \left( \boldsymbol{A}_k \mathbf{x} - \mathbf{b} \,;\, \mathbf{x} \right) \right\|_{2,\varepsilon} \pm \left( 1 + \Delta + \sqrt{\tfrac{2}{\delta}} \right) \left\| \left( \boldsymbol{A}_{\hat{k}}\hat{\mathbf{x}} - \mathbf{b} \,;\, \hat{\mathbf{x}} \right) \right\|_{2,\varepsilon}
\end{aligned}
$$

Note that the above bound holds for any choice of $\boldsymbol{A}_k$ and any $\mathbf{x}$. To simplify the constants a bit, let $c_0 := \left( 1 + \Delta + \sqrt{\tfrac{2}{\delta}} \right)$, $\mathcal{L}(k, \mathbf{x}) := \left\| (\boldsymbol{A}_k \mathbf{x} - \mathbf{b} \,;\, \mathbf{x}) \right\|_{2,\varepsilon}$, and $L(k, \mathbf{x}) := \left\| (\boldsymbol{S}(\boldsymbol{A}_k \mathbf{x} - \mathbf{b}) \,;\, \mathbf{x}) \right\|_{2,\varepsilon}$. Then, the previous bound state that

$$
L(k, \mathbf{x}) \in (1 \pm \Delta)\mathcal{L}(k, \mathbf{x}) \pm c_0 \mathcal{L}(\hat{k}, \hat{\mathbf{x}})
$$

If we take $k = \tilde{k}$ and $\mathbf{x} = \tilde{\mathbf{x}}$, and rearrange terms, we find

$$
\begin{aligned}
\mathcal{L}(\tilde{k}, \tilde{\mathbf{x}}) &\le \frac{1}{1 - \Delta} L(\tilde{k}, \tilde{\mathbf{x}}) + \frac{c_0}{1 - \Delta} \mathcal{L}(\hat{k}, \hat{\mathbf{x}}) \\
&\le \frac{1}{1 - \Delta} L(\hat{k}, \hat{\mathbf{x}}) + \frac{c_0}{1 - \Delta} \mathcal{L}(\hat{k}, \hat{\mathbf{x}}) \\
&\le \frac{1}{1 - \Delta} \left( (1 + \Delta)\mathcal{L}(\hat{k}, \hat{\mathbf{x}}) + c_0 \mathcal{L}(\hat{k}, \hat{\mathbf{x}}) \right) + \frac{c_0}{1 - \Delta} \mathcal{L}(\hat{k}, \hat{\mathbf{x}}) \\
&= \frac{1 + \Delta + 2c_0}{1 - \Delta} \mathcal{L}(\hat{k}, \hat{\mathbf{x}}) \\
&= \frac{1 + \Delta + 2\left( 1 + \Delta + \sqrt{\tfrac{2}{\delta}} \right)}{1 - \Delta} \mathcal{L}(\hat{k}, \hat{\mathbf{x}}) \\
&= \frac{3 + 3\Delta + 2\sqrt{\tfrac{2}{\delta}}}{1 - \Delta} \mathcal{L}(\hat{k}, \hat{\mathbf{x}})
\end{aligned}
$$

If we take $\Delta = \tfrac{1}{3}$ and square both sides, we find

$$
\|\boldsymbol{A}_{\tilde{k}}\tilde{\mathbf{x}} - \mathbf{b}\|_2^2 + \varepsilon\|\tilde{\mathbf{x}}\|_2^2 \le \left( 6 + 3\sqrt{\tfrac{2}{\delta}} \right)^2 \left( \|\boldsymbol{A}_{\hat{k}}\hat{\mathbf{x}} - \mathbf{b}\|_2^2 + \varepsilon\|\hat{\mathbf{x}}\|_2^2 \right)
$$

Noting by the AM-GM inequality that $(a + b)^2 = a^2 + 2ab + b^2 \le 2a^2 + 2b^2$, we can upper bound the above approximation factor with $\left( 6 + 3\sqrt{\tfrac{2}{\delta}} \right)^2 \le (72 + \tfrac{18}{\delta})$, completing the proof. $\qquad \square$

### C.2 Proof for the Operator Case

We start with preliminary definitions for randomized operator analysis.

#### C.2.1 Ridge Leverage Scores

To achieve near optimal sample complexity for kernel interpolation (i.e within logarithmic factors of the statistical dimension), recent work shows that it suffices to select time samples independently at random, according to a carefully chosen non-uniform distribution [CKPS16, AKM$^+$19]. In particular, we use the well studied ridge leverage function [AM15, MM17, PBV18], which is defined as follows:

**Definition 4** (Ridge leverage function). *For any Hilbert space $\mathcal{H}$, time length $T > 0$, $\varepsilon \ge 0$, and bounded linear operator $\mathcal{A} : \mathcal{H} \to L_2(T)$ the $\varepsilon$-ridge leverage function for $t \in [0, T]$ is:*

$$
\tau_{\mathcal{A},\varepsilon}(t) = \frac{1}{T} \cdot \max_{\{\alpha \in \mathcal{H} \,:\, \|\alpha\|_{\mathcal{H}} > 0\}} \frac{|[\mathcal{A}\alpha](t)|^2}{\|\mathcal{A}\alpha\|_T^2 + \varepsilon\|\alpha\|_{\mathcal{H}}^2}. \tag{7}
$$

Note that when $\mathcal{A}$ is an inverse Fourier transform operator, the integral of the ridge leverage function is equal to the statistical dimension of the corresponding kernel – i.e. if $\mathcal{A} = \mathcal{F}_\mu^*$ then $s_{\mu,\varepsilon} = \int_0^T \tau_{\mathcal{A},\varepsilon}(t)dt$. This fact generalizes a well known claim for matrices and is proven in [AKM$^+$19]. The ridge leverage score captures how important a time point $t$ is for $\mathcal{A}$: it is large if there are low energy functions (small $\|\alpha\|_{\mathcal{H}}^2$) in the span of the operator that are highly concentrated at $t$ – i.e. when the function $\mathcal{A}\alpha$ has large magnitude at $t$ compared to its average magnitude over $[0,T]$.

The Universal Sampling Distribution (Definition 2) is called *Universal* because when $\mathcal{A} = \mathcal{F}_\mu^*$ is *any* inverse Fourier transform operator, recent work [CP19a, AKM$^+$19] shows that $\tau_{\mathcal{A},\varepsilon}$ is tightly upper bounded by $\tilde{\tau}_\alpha$:

**Claim 2** (Theorem 17 of [AKM$^+$19]). *For any PDF $\mu$ and corresponding inverse Fourier transform operator $\mathcal{F}_\mu^*$,*

$$\tau_{\mathcal{F}_\mu^*,\varepsilon}(t) \leq \tilde{\tau}_\alpha(t)$$

*for all $t \in [0,T]$, as long as $\alpha \geq c s_{\mu,\varepsilon}$ for some universal constant $c > 0$.*

We then state a known operator subsampling result from [AKM$^+$19] which is based on the ridge leverage scores of Definition 4. The proof of this result adapts a bound on sums of random operators by [Min17], and uses the upper bound of Claim 2. A similar result is proven in [Bac17].

**Lemma 3** (Lemma 43 in [AKM$^+$19]). *Consider a bounded linear operator $\mathcal{A} : \mathcal{H} \to L_2(T)$. Let $\tilde{\tau}_{\mathcal{A},\varepsilon}(t)$ be a function with $\tilde{\tau}_{\mathcal{A},\varepsilon}(t) \geq \tau_{\mathcal{A},\varepsilon}(t)$ for all $t \in [0,T]$ and let $\tilde{s}_{\mathcal{A},\varepsilon} = \int_0^T \tilde{\tau}_{\mathcal{A},\varepsilon}(t)dt$. Let $n = c \cdot \Delta^{-2}\tilde{s}_{\mathcal{A},\varepsilon} \log(\tilde{s}_{\mathcal{A},\varepsilon}/\delta)$ for sufficiently large fixed constant $c$ and select $t_1, \ldots, t_n$ by drawing each randomly from $[0,T]$ with probability proportional to $\tilde{\tau}_{\mathcal{A},\varepsilon}(t)$. For $j \in 1, \ldots, s$, let $w_j = \sqrt{\frac{\tilde{s}_{\mathcal{A},\varepsilon}}{nT \cdot \tilde{\tau}_{\mathcal{A},\varepsilon}(t_j)}}$. Let $\mathbf{A} : \mathcal{H} \to \mathbb{C}^n$ be the operator defined by $[\mathbf{A}g]_j = [\mathcal{A}g](t_j) \cdot w_j$. With probability $(1 - \delta)$,*

$$(1 - \Delta)(\mathcal{G} + \varepsilon\mathcal{I}_{\mathcal{H}}) \preceq \mathbf{A}^*\mathbf{A} + \varepsilon\mathcal{I}_{\mathcal{H}} \preceq (1 + \Delta)(\mathcal{G} + \varepsilon\mathcal{I}_{\mathcal{H}}).$$

### C.2.2 Concentration of Concatenated Fourier Operators

With Lemma 3 in place, our goal in this section is prove a specific approximation result for randomly subsampling rows from the the *concatenation* of two inverse Fourier transform operators, $\mathcal{F}_{\mu_1}^*$ and $\mathcal{F}_{\mu_2}^*$. Specifically, let $\oplus$ denote the standard direct sum operation between Hilbert spaces. I.e. $[\alpha, \beta] \in \mathcal{H}_1 \oplus \mathcal{H}_2$ if $\alpha \in \mathcal{H}_1$ and $\beta \in \mathcal{H}_1$. For finitely bounded PDFs $\mu_1$ and $\mu_2$ the concatenated operator $\mathcal{F}_{\mu_1,\mu_2}^* : L_2(\mu_1) \oplus L_2(\mu_2) \to L_2(T)$ is defined as:

$$\mathcal{F}_{\mu_1,\mu_2}^*[\alpha, \beta] = \mathcal{F}_{\mu_1}^*\alpha + \mathcal{F}_{\mu_2}^*\beta.$$

Note that the adjoint of $\mathcal{F}_{\mu_1,\mu_2}^*$ is $\mathcal{F}_{\mu_1,\mu_2}f = (\mathcal{F}_{\mu_1}f, \mathcal{F}_{\mu_2}f)$.

Our goal is to approximate $\mathcal{F}_{\mu_1,\mu_2}^*$ by an operator with a finite number of rows. Such an approximation could be obtained directly from Lemma 3. However, applying that result requires *an upper bound on the ridge leverage scores (Definition 4) of $\mathcal{F}_{\mu_1,\mu_2}^*$*. Our first technical result of this section is to show that such an upper bound can be obtained using the universal sampling distribution of Definition 2. We prove:

**Lemma 4.** *For any bounded PDFs $\mu_1, \mu_2$ on $\mathbb{R}$ let, $\mathcal{A} = \mathcal{F}_{\mu_1,\mu_2}^*$ where $\mathcal{F}_{\mu_1,\mu_2}^*$ is a concatenated inverse Fourier transform operator as defined above for any $\varepsilon > 0$,*

$$\tau_{\mathcal{A},\varepsilon}(t) \leq \tilde{\tau}_\alpha(t)$$

*as long as $\alpha \geq c \cdot \max[s_{\mu_1,\varepsilon}, s_{\mu_2,\varepsilon}]$ for some fixed constant $c$.*

*Proof.* Let $\bar{\mu} = \frac{\mu_1 + \mu_2}{2}$ and let $\bar{\mathcal{A}} = 2\mathcal{F}_{\bar{\mu}}^*$. We establish the lemma by proving

$$\tau_{\mathcal{A},\varepsilon}(t) \leq \tau_{\bar{\mathcal{A}},\varepsilon}(t) \tag{8}$$

Once we have this bound, we can apply Claim 2 to observe that $\tau_{\bar{\mathcal{A}},\varepsilon}(t) \leq \tilde{\tau}_\alpha(t)$ as long as long as $\alpha \geq c \cdot s_{\bar{\mu},\varepsilon}$. Finally, from Lemma 51 in [AKM$^+$19], we have that $s_{\bar{\mu},\varepsilon} \leq 2\max[s_{\mu_1,\varepsilon}, s_{\mu_2,\varepsilon}]$, which gives the lemma because $\tilde{\tau}_\alpha(t)$ is strictly increasing with $\alpha$.

So, we are left to prove (8). Referring to Definition 4 and noting that $\|[\alpha,\beta]\|^2_{\mathcal{H}_1 \oplus \mathcal{H}_2} = \|\alpha\|^2_{\mathcal{H}_1} + \|\beta\|^2_{\mathcal{H}_2}$, we can do so by upper bounding for all $t \in [0,T]$:

$$\frac{1}{T} \cdot \max_{\{[\alpha,\beta] \in L_2(\mu_1) \oplus L_2(\mu_2): \|\alpha\|^2_{\mu_1} + \|\beta\|^2_{\mu_2} > 0\}} \frac{\left|[\mathcal{F}^*_{\mu_1,\mu_2}[\alpha,\beta]](t)\right|^2}{\|\mathcal{F}_{\mu_1,\mu_2}[\alpha,\beta]\|^2_T + \varepsilon\|\alpha\|^2_{\mu_1} + \varepsilon\|\beta\|^2_{\mu_2}}. \tag{9}$$

For any particular $t \in [0,T]$, let $\alpha^* \in L_2(\mu_1)$ and $\beta^* \in L_2(\mu_1)$ be the maximizers of (9). We are going to define a function $w$ to satisfy $\bar{A}w = \mathcal{F}^*_{\mu_1,\mu_2}[\alpha^*,\beta^*]$. In particular, we can set

$$w(\xi) = (\mu_1(\xi) + \mu_2(\xi))^+ (\mu_1(\xi)\alpha^*(\xi) + \mu_2(\xi)\beta^*(\xi))$$

where for a $s \in \mathbb{R}$, $s^+$ evaluates to $0$ when $s = 0$ and $1/s$ otherwise. We have that (9) is equal to:

$$\frac{1}{T} \cdot \frac{\left|[\bar{A}w](t)\right|^2}{\|\bar{A}w\|^2_T + \varepsilon\|\alpha^*\|^2_{\mu_1} + \varepsilon\|\beta^*\|^2_{\mu_2}}. \tag{10}$$

Next we bound $\|w\|^2_{\bar{\mu}} = \int_{\xi \in \mathbb{R}} w(\xi)^2 \bar{\mu}(\xi) d\xi$. We have that for all $\xi$,

$$w(\xi)^2 \bar{\mu} = \frac{1}{2}(\mu_1(\xi) + \mu_2(\xi))^+ \cdot (\mu_1(\xi)\alpha^*(\xi) + \mu_2(\xi)\beta^*(\xi))^2$$
$$\leq (\mu_1(\xi) + \mu_2(\xi))^+ \cdot \left(\mu_1(\xi)^2\alpha^*(\xi)^2 + \mu_2(\xi)^2\beta^*(\xi)^2\right)$$
$$\leq \mu_1(\xi)\alpha^*(\xi)^2 + \mu_2(\xi)\beta^*(\xi)^2.$$

It follows that $\int_{\xi \in \mathbb{R}} w(\xi)^2 \bar{\mu}(\xi) d\xi \leq \int_{\xi \in \mathbb{R}} \alpha^*(\xi)^2 \mu_1(\xi) d\xi + \int_{\xi \in \mathbb{R}} \beta^*(\xi)^2 \mu_2(\xi) d\xi = \|\alpha^*\|^2_{\mu_1} + \|\beta^*\|^2_{\mu_2}$. Substituting into (10), we actually have that (9) can be upper bounded by

$$\frac{1}{T} \cdot \frac{\left|[\bar{A}w](t)\right|^2}{\|\bar{A}w\|^2_T + \varepsilon\|w\|^2_{\bar{\mu}}}.$$

This quantity is of course only small than $\tau_{\bar{A},\varepsilon}(t)$, which completes the proof of (8). $\square$

The following theorem is a direct corollary of Lemma 3 and Lemma 4.

**Theorem 6.** *Fix $\Delta > 0$ and $\delta > 0$. Let $\mu_1, \mu_2$ be bounded PDFs. Let $s_{max} = \max[s_{\mu_1,\varepsilon}, s_{\mu_2,\varepsilon}]$. Let $\alpha = c_0 s_{max}$ and $n = c_1 \Delta^{-2} s_{max} \log(s_{max}) \log(s_{max}/\delta)$ for fixed universal constants $c_0, c_1$. Suppose $n$ time samples $t_1, \ldots, t_n \in [0,T]$ are sampled with probability proportional to $\tilde{\tau}_\alpha(t)$ and $\boldsymbol{F}^*_{\mu_1}$ and $\boldsymbol{F}^*_{\mu_2}$ be the sampled versions of $\mathcal{F}^*_{\mu_1}$ and $\mathcal{F}^*_{\mu_2}$ satisfying for $j = 1, \ldots, n$:*

$$[\boldsymbol{F}^*_{\mu_p}g]_j = w_j \cdot \int_{\mathbb{R}} g(\xi)e^{2\pi i \xi t_j} \mu_p(\xi) d\xi,$$

*where $w_j = \sqrt{\frac{\int_0^T \tilde{\tau}_\alpha(t)dt}{sT \cdot \tilde{\tau}_\alpha(t_j)}}$. Then with probability $(1 - \delta)$,*

$$(1 - \Delta)(\mathcal{G} + \varepsilon\mathcal{I}) \preceq \tilde{\mathcal{G}} + \varepsilon\mathcal{I} \preceq (1 + \Delta)(\mathcal{G} + \varepsilon\mathcal{I})$$

*where $\mathcal{G} = \mathcal{F}_{\mu_1,\mu_2}\mathcal{F}^*_{\mu_1,\mu_2}$ and $\bar{\mathcal{G}} = [\boldsymbol{F}_{\mu_2}; \boldsymbol{F}_{\mu_1}][\boldsymbol{F}^*_{\mu_2}, \boldsymbol{F}^*_{\mu_1}]$. Here $[\boldsymbol{F}_{\mu_2}; \boldsymbol{F}_{\mu_1}] : \mathbb{C}^s \to L_2(\mu_1) \oplus L_2(\mu_2)$ is the natural concatenation of $\boldsymbol{F}_{\mu_2}$ and $\boldsymbol{F}_{\mu_1}$, and $[\boldsymbol{F}^*_{\mu_2}, \boldsymbol{F}^*_{\mu_1}]$ is the concatenation of $\boldsymbol{F}^*_{\mu_2}$ and $\boldsymbol{F}^*_{\mu_1}$.*

*Proof.* By Lemma 4 $\tilde{\tau}_\alpha(t)$ strictly upper bounds the $\varepsilon$-ridge leverage scores of $\mathcal{F}^*_{\mu_1,\mu_2}$ as long as $\alpha$ is set as in the theorem statement. Moreover, referring to Definition 2, $\int_0^T \tilde{\tau}_\alpha(t)dt \leq O(\alpha \log \alpha)$, so the number of samples $n$ in the theorem is sufficiently large to directly apply Lemma 3 to the bounded linear operator $\mathcal{F}^*_{\mu_1,\mu_2}$. $\square$

### C.2.3 Final Result for Linear Operators

**Theorem 1 Restated.** *Let $\tilde{\mathcal{U}} = \{\mu_1, \ldots, \mu_Q\}$ be a finite set of scaled PDFs. Let $s_{max,\varepsilon}$ be the maximum of the PDFs' statistical dimensions: $s_\varepsilon = \max_j s_{\mu_j,\varepsilon}$. Let $t_1, \ldots, t_n$ be iid samples from the universal sampling distribution, and define $\boldsymbol{F}^*$ accordingly. Let $\tilde{\mu}, \tilde{g}$ optimally solve the time-discretized problem:*

$$\tilde{\mu}, \tilde{g} := \underset{\mu \in \tilde{\mathcal{U}}, g \in L_2(\mu)}{\operatorname{argmin}} \|\boldsymbol{F}^*_{\tilde{\mu}} g - \bar{\mathbf{y}}\|_2^2 + \varepsilon \|g\|_\mu^2$$

*If $n = O(s_\varepsilon \log(\frac{s_\varepsilon Q}{\delta}))$, then with probability $1 - \delta$, we have*

$$\|\mathcal{F}^*_{\tilde{\mu}} \tilde{g} - \bar{y}\|_T^2 + \varepsilon \|\tilde{g}\|_{\tilde{\mu}}^2 \leq (72 + \tfrac{18}{\delta}) \underset{\mu \in \tilde{\mathcal{U}}, g \in L_2(\mu)}{\operatorname{argmin}} \|\mathcal{F}^*_\mu g - \bar{y}\|_T^2 + \varepsilon \|g\|_\mu^2$$

*Proof.* Like in the proof of Theorem 5, we define regularized norms for this problem. However, since the regularization norm depends on the PDF being used, we use notionally larger norms:

$$\|(f\,;g\,;g')\|_{T,\varepsilon,\mu,\mu'}^2 := \|f\|_T^2 + \varepsilon\|g\|_\mu^2 + \varepsilon\|g'\|_{\mu'}^2 \qquad \|(f\,;g)\|_{T,\varepsilon,\mu}^2 := \|f\|_T^2 + \varepsilon\|g\|_\mu^2$$
$$\|(\mathbf{x}\,;g\,;g')\|_{2,\varepsilon,\mu,\mu'}^2 := \|\mathbf{x}\|_2^2 + \varepsilon\|g\|_\mu^2 + \varepsilon\|g'\|_{\mu'}^2 \qquad \|(\mathbf{x}\,;g)\|_{2,\varepsilon,\mu}^2 := \|\mathbf{x}\|_2^2 + \varepsilon\|g\|_\mu^2$$

Then let $\hat{\mu}$ and $\hat{\mathbf{x}}$ be the true minimizers for the full optimization problem:

$$\hat{\mu}, \hat{\mathbf{x}} := \underset{\mu \in \tilde{\mathcal{U}}, g \in L_2(\mu)}{\operatorname{argmin}} \|(\mathcal{F}^*_\mu g - \bar{y}\,;g)\|_{T,\varepsilon,\mu}^2$$

By the triangle inequality, and the inverse triangle inequality, we have for any $\mu$ and any $g \in L_2(\mu)$,

$$\|(\boldsymbol{F}^*_\mu g - \bar{\mathbf{y}}\,;g)\|_{2,\varepsilon,\mu} = \|(\boldsymbol{F}^*_\mu g - \bar{\mathbf{y}}\,;g\,;0)\|_{2,\varepsilon,\mu,\hat{\mu}}$$
$$\in \left\|\left(\boldsymbol{F}^*_{\hat{\mu}} \hat{g} - \boldsymbol{F}^*_\mu g\,;-g\,;\hat{g}\right)\right\|_{2,\varepsilon,\mu,\hat{\mu}} \pm \left\|\left(\boldsymbol{F}^*_{\hat{\mu}} \hat{g} - \bar{\mathbf{y}}\,;0\,;\hat{g}\right)\right\|_{2,\varepsilon,\mu,\hat{\mu}}$$

We bound these two terms separately, starting with the latter. Note from [AKM$^+$19] that $\mathbb{E}[\|\boldsymbol{F}^*_{\hat{\mu}} \hat{g} - \bar{\mathbf{y}}\|_2^2] = \|\mathcal{F}^*_{\hat{\mu}} \hat{g} - \bar{y}\|_T^2$. Hence, by Markov's inequality, we have

$$\left\|\left(\boldsymbol{F}^*_{\hat{\mu}} \hat{g} - \bar{\mathbf{y}}\,;0\,;\hat{g}\right)\right\|_{2,\varepsilon,\mu,\hat{\mu}} \leq \sqrt{\frac{2}{\delta}} \left\|\left(\mathcal{F}^*_{\hat{\mu}} \hat{g} - \bar{y}\,;0\,;\hat{g}\right)\right\|_{T,\varepsilon,\mu,\hat{\mu}}$$
$$= \sqrt{\frac{2}{\delta}} \left\|\left(\mathcal{F}^*_{\hat{\mu}} \hat{g} - \bar{y}\,;\hat{g}\right)\right\|_{T,\varepsilon,\hat{\mu}}$$

with probability $1 - \frac{\delta}{2}$.

Next, note that for constant $\Delta$ and for any $\mu_j, \mu_k \in \tilde{\mathcal{U}}$ we have

$$\|(\boldsymbol{F}^*_{\mu_j} g + \boldsymbol{F}^*_{\mu_k} h\,;g\,;h)\|_{2,\varepsilon,\mu_j,\mu_k} \in (1 \pm \Delta) \|(\mathcal{F}^*_{\mu_j} g + \mathcal{F}^*_{\mu_k} h\,;g\,;h)\|_{T,\varepsilon,\mu_j,\mu_k}$$

for all $g \in L_2(\mu_j), h \in L_2(\mu_k)$ with probability $1 - \frac{\delta}{2}$. This follows directly from Theorem 6.

Further, by the triangle and inverse inequalities, we have for any $\mu \in \tilde{\mathcal{U}}, g \in L_2(\mu)$

$$\left\|\left(\mathcal{F}^*_{\hat{\mu}} \hat{g} - \mathcal{F}^*_\mu g\,;-g\,;\hat{g}\right)\right\|_{T,\varepsilon,\mu,\hat{\mu}} \in \left\|\left(\mathcal{F}^*_\mu g - \bar{y}\,;g\,;0\right)\right\|_{T,\varepsilon,\mu,\hat{\mu}} \pm \left\|\left(\mathcal{F}^*_{\hat{\mu}} \hat{g} - \bar{y}\,;0\,;\hat{g}\right)\right\|_{T,\varepsilon,\mu,\hat{\mu}}$$
$$= \left\|\left(\mathcal{F}^*_\mu g - \bar{y}\,;g\right)\right\|_{T,\varepsilon,\mu} \qquad \pm \left\|\left(\mathcal{F}^*_{\hat{\mu}} \hat{g} - \bar{y}\,;\hat{g}\right)\right\|_{T,\varepsilon,\hat{\mu}}$$

Putting these last two inequalities together, we find

$$\left\|\left(\boldsymbol{F}^*_{\hat{\mu}} \hat{g} - \boldsymbol{F}^*_\mu g\,;-g\,;\hat{g}\right)\right\|_{2,\varepsilon,\mu,\hat{\mu}} \in (1 \pm \Delta)\left\|\left(\mathcal{F}^*_\mu g - \bar{y}\,;g\right)\right\|_{T,\varepsilon,\mu} \pm (1 + \Delta)\left\|\left(\mathcal{F}^*_{\hat{\mu}} \hat{g} - \bar{y}\,;\hat{g}\right)\right\|_{T,\varepsilon,\hat{\mu}}$$

Then, using this bound, alongside the Markov bound and the original triangle inequalities, we find

$$\|(\boldsymbol{F}^*_\mu g - \bar{\mathbf{y}}\,;g)\|_{2,\varepsilon,\mu} \in \left\|\left(\boldsymbol{F}^*_{\hat{\mu}} \hat{g} - \boldsymbol{F}^*_\mu g\,;-g\,;\hat{g}\right)\right\|_{2,\varepsilon,\mu,\hat{\mu}} \pm \left\|\left(\boldsymbol{F}^*_{\hat{\mu}} \hat{g} - \bar{\mathbf{y}}\,;0\,;\hat{g}\right)\right\|_{2,\varepsilon,\mu,\hat{\mu}}$$

$$\subseteq \left\|\left(\boldsymbol{F}^*_{\hat{\mu}} \hat{g} - \boldsymbol{F}^*_\mu g\,;-g\,;\hat{g}\right)\right\|_{2,\varepsilon,\mu,\hat{\mu}} \pm \sqrt{\tfrac{2}{\delta}} \left\|\left(\mathcal{F}^*_{\hat{\mu}} \hat{g} - \bar{y}\,;\hat{g}\right)\right\|_{T,\varepsilon,\hat{\mu}}$$

$$\subseteq \left((1 \pm \Delta)\left\|\left(\mathcal{F}^*_\mu g - \bar{y}\,;g\right)\right\|_{T,\varepsilon,\mu} \pm (1 + \Delta)\left\|\left(\mathcal{F}^*_{\hat{\mu}} \hat{g} - \bar{y}\,;\hat{g}\right)\right\|_{T,\varepsilon,\hat{\mu}}\right) \pm \sqrt{\tfrac{2}{\delta}} \left\|\left(\mathcal{F}^*_{\hat{\mu}} \hat{g} - \bar{y}\,;\hat{g}\right)\right\|_{T,\varepsilon,\hat{\mu}}$$

$$= (1 \pm \Delta)\left\|\left(\mathcal{F}^*_\mu g - \bar{y}\,;g\right)\right\|_{T,\varepsilon,\mu} \pm \left(1 + \Delta + \sqrt{\tfrac{2}{\delta}}\right) \left\|\left(\mathcal{F}^*_{\hat{\mu}} \hat{g} - \bar{y}\,;\hat{g}\right)\right\|_{T,\varepsilon,\hat{\mu}}$$

To simplify the notation a bit, let $c_0 := \left(1 + \Delta + \sqrt{\frac{2}{\delta}}\right)$, the true loss for PDF $\mu$ with signal $g$ be $\mathcal{L}(\mu, g) := \left\| \left(\mathcal{F}_\mu^* g - \bar{y} \, ; \, g\right) \right\|_{T, \varepsilon, \mu}$, and the sample loss for PDF $\mu$ with signal $g$ be $L(\mu, g) := \| (\boldsymbol{F}_\mu^* g - \bar{\mathbf{y}} \, ; \, g) \|_{2, \varepsilon, \mu}$. Then the previous bound says

$$L(\mu, g) \in (1 \pm \Delta)\mathcal{L}(\mu, g) \pm c_0 \mathcal{L}(\hat{\mu}, \hat{g})$$

Recall that this bound holds for any choice of $\mu \in \tilde{\mathcal{U}}$ and any $g \in L_2(\mu)$. If we take $\mu = \tilde{\mu}$ and $g = \tilde{g}$, and rearrange terms, we find

$$
\begin{aligned}
\mathcal{L}(\tilde{\mu}, \tilde{g}) &\leq \frac{1}{1 - \Delta} L(\tilde{\mu}, \tilde{g}) + \frac{c_0}{1 - \Delta} \mathcal{L}(\hat{\mu}, \hat{g}) \\
&\leq \frac{1}{1 - \Delta} L(\hat{\mu}, \hat{g}) + \frac{c_0}{1 - \Delta} \mathcal{L}(\hat{\mu}, \hat{g}) \\
&\leq \frac{1}{1 - \Delta} \left((1 + \Delta)\mathcal{L}(\hat{\mu}, \hat{g}) + c_0 \mathcal{L}(\hat{\mu}, \hat{g})\right) + \frac{c_0}{1 - \Delta} \mathcal{L}(\hat{\mu}, \hat{g}) \\
&= \frac{1 + \Delta + 2c_0}{1 - \Delta} \mathcal{L}(\hat{\mu}, \hat{g}) \\
&= \frac{1 + \Delta + 2(1 + \Delta + \sqrt{\frac{2}{\delta}})}{1 - \Delta} \mathcal{L}(\hat{\mu}, \hat{g}) \\
&= \frac{3 + 3\Delta + 2\sqrt{\frac{2}{\delta}}}{1 - \Delta} \mathcal{L}(\hat{\mu}, \hat{g})
\end{aligned}
$$

If we take $\Delta = \frac{1}{3}$, we find

$$(\mathcal{L}(\tilde{\mu}, \tilde{g}))^2 \leq \left(6 + 3\sqrt{\frac{2}{\delta}}\right)^2 (\mathcal{L}(\hat{\mu}, \hat{g}))^2$$

$$\|\mathcal{F}_{\tilde{\mu}} \tilde{g} - \bar{y}\|_T^2 + \varepsilon \|\tilde{g}\|_{\tilde{\mu}}^2 \leq \left(6 + 3\sqrt{\frac{2}{\delta}}\right)^2 \left(\|\mathcal{F}_{\hat{\mu}} \hat{g} - \bar{y}\|_T^2 + \varepsilon \|\hat{g}\|_{\hat{\mu}}^2\right)$$

Noting by the AM-GM inequality that $(a + b)^2 = a^2 + 2ab + b^2 \leq 2a^2 + 2b^2$, we can upper bound the above approximation factor with $\left(6 + 3\sqrt{\frac{2}{\delta}}\right)^2 \leq (72 + \frac{18}{\delta})$, completing the proof. $\qquad \square$