[Reviews · NeurIPS 2020]

Review 1

Summary and Contributions: This submission proves that the sample complexity of learning kernel hyperparameters under adversarial noise is not much larger than the sample complexity for the setting with known optimal parameters. This work not only has theoretical contributions, but also can inspire practitioners to develop more effective algorithms to tune Spectral Mixture kernel in the future. Overall, I think this is a solid work.

Strengths: This work is technically sound. The conclusion of this submission reveals that the key to robust kernel hyperparameter tuning is not a larger amount of data, but the development of more effective algorithms. It makes theoretical contributions and provides guidance to practitioners.

Weaknesses: This work is an extension of [AKM+19].The novelty and impact of this work are unclear to me since I am not familiar with this field.

Correctness: Yes, I think this method is technically sound.

Clarity: Yes, this submission is well written and well organized.

Relation to Prior Work: Yes, the authors clearly discussed the related works. To my understand, this work is an extension of [AKM+19].

Reproducibility: Yes

Additional Feedback: As authors discussed in the conclusion section, this submission does not provide any polynomial time algorithm to solve the problem. It would be nice if the authors could discuss the possible direction of developing the corresponding algorithm in this paper. Thank you for authors to address my concerns in the rebuttal. I remain my score unchanged since authors' responses do not chance my evaluation of the overall quality of this submission.


Review 2

Summary and Contributions: This is a theoretical paper that analyzes the statistical cost of hyperparameter tuning. The authors give a background to previous work and present a series of theoretical results. === Post-rebuttal update === Thank you for your response! I've decided to keep my score as I'm still concerned about the broader impact and potential applications of this work. While I understand that this is a theoretical paper, I would still like to see comparisons to other methods to get a better understanding of its impact.

Strengths: I'm not familiar with this area of research, but the paper considers an interesting problem and is very well-written and seems theoretically sound.

Weaknesses: I'm not convinced that this work carries much practical importance and there are no numerical experiments in the paper. In fact, the authors say "This is a theoretical paper, so discussion of broader impacts has to speculate on future applications for this broad line of research.", which makes it sound like it isn't clear what potential applications are and the lack of experiments indicate that there were no clear applications to study. Other questions and comments: - Can you be a bit more explicit about what it means to use O(1) in the bounds? I understand it as a positive constant that doesn't depend on y, z, etc., but being explicit would be helpful. Also, in Problem 2 restated, state the dependencies on C. - Giving the reader some intuition of what energy means when Problem 1 and 2 are introduced will avoid jumping between the sections to understand what is going on. Drawing the connection to inner products in the RKHS would be useful too. - What is the intuition behind the statistical dimension and how does it vary between some of the commonly used stationary kernels (RBF, Matern, etc.)?

Correctness: I believe the claims are correct

Clarity: Yes

Relation to Prior Work: Yes

Reproducibility: Yes

Additional Feedback:


Review 3

Summary and Contributions: Within a Kernel Ridge Regression setting, this paper analyses the problem of interpolating a signal with adversarial noise to any arbitrary precision while learning the kernel hyperparameters. The paper proves that this problem can be solved with high probability if the observations positions are drawn from a kernel independent distribution and the number of samples exceeds a log-linear bound on the statistical dimension of the kernel family. These bounds, for the case of the Spectral Mixture kernel (SM), and the universal sampling distribution are the main contribution of the paper. For this, the paper generalizes the results of [1] by removing the fix kernel assumption and replacing it with a parametrized infinite kernel space. Similar to [1], the proof is done by showing first that the active regression problem with adversarial noise can be solved by finding a near-optimal solution to Fourier Fitting (FF) problem with ridge regularization. Then, the paper proves the existence of near-optimal solutions to the FF problem by first reducing it to the case where the hyperparameter space is finite and showing that, when observation positions are sampled with a kernel independent distribution, a solution for the FF problem exists with high probability. The paper then proposes a discretization scheme for the infinite hyperparameters space case. [1] Haim Avron, Michael Kapralov, Cameron Musco, Christopher Musco, Ameya Velingker, and Amir Zandieh. A universal sampling method for reconstructing signals with simple Fourier transforms. In Proceedings of the 51st Annual ACM SIGACT Symposium on Theory of Computing, pages 1051–1063. ACM, 2019.

Strengths: The main strength of the paper are the explicit bounds on the number of samples needed for achieving arbitrary low error when using the SM kernel, which could potentially give insights on improving the training of spectral kernels in general. This is relevant since spectral kernels are notoriously difficult to train but also of interest to the kernel community due to their universal approximation properties.

Weaknesses: Lack of experimental validation, probably due, as the paper mentions, a missing algorithm for solving the discretized hyperparameter space problem.

Correctness: The paper is technically correct.

Clarity: The presentation of the paper very clear and the outline of the proof is well presented.

Relation to Prior Work: The paper outlines clearly its differences with previous work.

Reproducibility: Yes

Additional Feedback: Post rebuttal: Post rebuttal: While the paper is promising and lays out the theoretical framework needed for tackling the complexity of fitting a SM kernel, its broader applications are still unclear. Thus, I am keeping the score as it is.


Review 4

Summary and Contributions: This paper considered the kernel hyperparameter tuning of Gaussian Mixture kernel in active regression with an eps-net style analysis based on a result in [AKM+19]. The authors showed polynomial sample complexity w.r.t number of Gaussian component, Gaussian length-scales and input range, and claimed the sample complexity is tight up to logarithm factor. Compared with given hyper-parameter, hyper-parameter tuning only requires an additional # of Gaussian components multiplicative factor if we ignore the logarithmic factor, which shows that hyper-parameter tuning only need an additional multiplicative factor samples under active regression setting.

Strengths: Theoretical results show that under active regression setting hyper-parameter tuning is not much harder than learning with given hyper-parameter.

Weaknesses: As the authors admitted, the time complexity is still exponential w.r.t $ of Gaussian components, as the current algorithm need to enumerate all of the combinations of hyper-parameters.

Correctness: The derivations are correct or easy to correct without technical issue if the previous results in [AKM+19] are correct.

Clarity: The presentation is clear.

Relation to Prior Work: The authors discussed the relation with prior work clearly.

Reproducibility: Yes

Additional Feedback: Overall the idea is clear and easy to follow. The authors focused on the specific setting of active regression with Gaussian Mixture kernel and demonstrate that we don’t need much more sample compared with learning with pre-fixed hyperparameter, which is a satisfactory result if we focus on active regression setting. One of my main concern, as the authors pointed out, is the practical algorithm when we need to use multiple Gaussian components. And I’m curious about if the techniques can be generalized to the setting with fixed data, which is more general and also indicates some fundamental limit in the kernel learning.

[Author Response · NeurIPS 2020]

**Recap: What is the paper's goal, and why?** Bound the statistical complexity of tuning SM Kernel hyperparameters,
when **noise is fully adversarial** in the **active regression** setting. While extremely useful, the SM Kernel is notoriously
hard to tune (L147-148). By proving that the statistical cost of tuning the SM Kernel is low, even in this hard regime,
we show that we need better algorithms for hyperparameter tuning, and not to just collect more data. Also, we provide a
new mathematical framework to analyze hyperparameter tuning, which hopefully opens the door to new algorithms.

We thank the reviewers for their constructive and broadly positive feedback. We are pleased to see the reviewers find
our problem statement interesting and relevant, find our technical results to be sound and relevant to practitioners, and
find our writing to be clear. We also appreciate the reviewers asking about potential future directions of our work, either
for new algorithms or for a different statistical setup.

One important open question identified by (R1, R3, R4) is about **finding a polynomial time algorithm**. While
our paper does not have a polynomial time algorithm for SM Kernel hyperparameter tuning, **it opens the door for**
**more research towards that goal**. Previously, it was unknown if exponentially many samples were needed to even
*information-theoretically* tune a SM Kernel. If this were the case, then there would be no hope for finding a polynomial
time algorithm. We show that this barrier does not exist, which gives more hope for a fast algorithm.

Along the same lines, R1 asks us to **"discuss the possible direction[s] of developing the corresponding algorithm"**.
This highlights another potential benefit of our paper: prior work does not frame hyperparameter tuning as a Fourier
fitting problem. By rigorously and directly connecting the two settings, our paper makes it easier for a polynomial
time algorithm to originate from signal processing and benefit kernel hyperparameter tuning. **The Sparse Fourier**
**Transform and Compressed Sensing literatures seem promising in this regard** – notably, they both feature fast
algorithms with statistical guarantees for many Fourier fitting problems. We agree that the paper should discuss these
directions explicitly, and we have added such a discussion to our conclusion.

R2 is concerned about the **applications and broader impact of our paper**. We study kernel ridge regression with
adversarial noise. There are many applications of this setting. For instance, we cite two papers that use the SM Kernel in
distinct ways: [HSSM15] uses the SM Kernel in an analysis of the lifespan of lithium-ion batteries and [WDLX15] uses
the SM Kernel to model human decision-making processes. Many more applications can by found in the papers that
cite [WA13], which proposed the SM Kernel. So, **the SM Kernel is relevant to many applications** despite being hard
to tune. Our **broader impact depends on how the SM Kernel is used**. We agree the introduction and broader impact
sections would benefit from a clearer discussion of applications of our framework, and have added that to the paper.

R1 also brings up the **novelty and impact of our paper, and its relationship to [AKM+19]**. [AKM+19] is a recent
STOC paper about Fourier function fitting in the same adversarial noise setting as our paper. They show that kernel
ridge regression provides a good interpolant when the kernel is fixed (i.e. hyperparameters are *known*). The novel
technical contribution of our work is the extension of [AKM+19] to the setting with an *unknown* kernel. This allows us
to frame kernel hyperparameter optimization as a Fourier fitting problem, so **[AKM+19] lets us prove the first bounds**
**on the statistical complexity of learning kernels with totally adversarial noise**. Mathematically, this comparison is
made clear in our introduction (Problem 1 versus Problem 2).

R4 asks **if our results can apply to the fixed data setting**. This is an interesting question, but answering it requires a
different noise model than our totally adversarial noise model. If the data was fixed, then an adversary could arbitrarily
perturb the signal $y(t)$ only at the observed times. So, a small-norm noise signal would make approximate recovery of
$y(t)$ totally impossible. Instead, perhaps the adversary could only perturb an $\varepsilon$ fraction of the observations? We think
this is an interesting direction, and it is plausible that our techniques could generalize such a fixed-kernel result into a
hyperparameter optimization result, but this speculation and out of scope for our paper.

R2 asks for a **"intuition behind the statistical dimension"** and how it varies between kernels. Kernels are nonpara-
metric functions that can involve infinitely many features (in our paper, a feature of a kernel is like the Fourier transform
of a kernel at a given frequency). Nevertheless, in Kernel Ridge Regression, thanks in part to regularization, kernels can
typically be well approximated with a finite number of features (this is the idea behind e.g. Random Fourier Features).
Statistical dimension captures exactly how many features are needed. Since regression in $d$ features requires roughly $d$
samples, this means kernel ridge regression needs roughly statistical dimension many observations. For the RBF kernel,
statistical dimension is linear in the lengthscale parameter and the duration of time we interpolate over. The intuition is
that if either the lengthscale or the duration of time increase, then we can represent more complex functions with the
same Fourier-norm, so this necessitates more samples (i.e., larger statistical dimension).

R2 asks for **Clarification about** $O(1)$ **and** $C$ **in our bounds**. These all refer to a universal constant independent of
any problem parameters such that the left hand side is bounded by the constant times the right hand side. We thank R2,
have updated our paper to always use the $C$ notation, and always describe $C$ as a universal constant. R2 asks us to
include a **better intuition for the Energy term in the introduction**, beyond what is on lines 33-35. We thank R2 and
agree; we have now included a stronger mathematical intuition for the Energy term in the introduction.

[Meta-Review · NeurIPS 2020]

This paper is concerned with active regression under adversarial noise, where the underlying function is assumed to live in a space induced by a parametric kernel with unknown hyperparameters (e.g., bandwidth of RBF kernel). Previous work shows that shows that if the kernel parameters are known exactly a priori, then efficient learning is possible. This work shows that if one does not know the kernel parameters a priori for a class of kernels, there exists an algorithm that learns these hyperparameters online and requires a sample complexity not much larger than if the kernel had been known for this adversarial active regression problem. However, while the statistical sample complexity is small, the computational complexity remains an issue and little guidance is given on how to overcome the exponential time naive algorithm. Moreover, reviewers also questioned the applicability of the approach—when is adversarial noise a good model? Where is this used? The authors addressed the applicability point in their rebuttal by simply restating a paper they had already cited, and not justified why adversarial noise was the right model here. The authors are strongly encouraged to make a case for how this work could potentially have impact in practice. Or on the other hand, acknowledge this is a theoretical result only and argue why it is fundamental.